# α-Melanocyte-stimulating hormone alleviates pathological cardiac remodeling via melanocortin 5 receptor

Anni Suominen[1,2], Guillem Saldo Rubio [1], Saku Ruohonen [1], Zoltán Szabó [3], Lotta Pohjolainen[4], Bishwa Ghimire [5,6], Suvi T Ruohonen[1], Karla Saukkonen[1], Jani Ijas[1], Sini Skarp [3], Leena Kaikkonen [3], Minying Cai [7], Sharon L Wardlaw [8], Heikki Ruskoaho[4], Virpi Talman [4], Eriika Savontaus[1,9,10], Risto Kerkelä [3,11,12] & Petteri Rinne [1,9]✉

## Abstract

α-Melanocyte-stimulating hormone (α-MSH) regulates diverse physiological functions by activating melanocortin receptors (MC-R). However, the role of α-MSH and its possible target receptors in the heart remain completely unknown. Here we investigate whether α-MSH could be involved in pathological cardiac remodeling. We found that α-MSH was highly expressed in the mouse heart with reduced ventricular levels after transverse aortic constriction (TAC). Administration of a stable α-MSH analog protected mice against TAC-induced cardiac hypertrophy and systolic dysfunction. In vitro experiments revealed that MC5-R in cardiomyocytes mediates the anti-hypertrophic signaling of α-MSH. Silencing of MC5-R in cardiomyocytes induced hypertrophy and fibrosis markers in vitro and aggravated TAC-induced cardiac hypertrophy and fibrosis in vivo. Conversely, pharmacological activation of MC5-R improved systolic function and reduced cardiac fibrosis in TAC-operated mice. In conclusion, α-MSH is expressed in the heart and protects against pathological cardiac remodeling by activating MC5-R in cardiomyocytes. These results suggest that analogs of naturally occurring α-MSH, that have been recently approved for clinical use and have agonistic activity at MC5-R, may be of benefit in treating heart failure.

**Keywords** Melanocyte-stimulating Hormone; Melanocortin Receptor; Hypertrophy; Heart Failure; Fibrosis
**Subject Categories** Cardiovascular System; Molecular Biology of Disease; Signal Transduction

## Introduction

Heart failure is one of the leading causes of hospitalization, (Jessup and Brozena et al, 2003; Tsao et al, 2022; Ambrosy et al, 2014) causing enormous burden on our economic and health care systems. Despite currently available therapeutics and advancements in the clinical management of heart failure and its associated complications, many challenges still remain as morbidity and mortality rates are showing no signs of decrease, (Tsao et al, 2022) highlighting unmet medical need to develop novel disease-modifying therapies for this disease.

Heart failure is a common end-stage manifestation for many cardiovascular diseases and it primarily results from hypertension, ischemic heart disease and aortic valve stenosis. Hypertension, for instance, increases cardiac workload and induces left ventricular (LV) hypertrophy, which is compensatory and adaptive in nature and helps to maintain circulatory homeostasis. However, in the long term, the stressed heart undergoes maladaptive remodeling resulting in the development of fibrosis, LV dilatation, and dysfunction, thus predisposing individuals to heart failure (Tham et al, 2015; Van Berlo et al, 2013). Melanocortins are peptide hormones that are proteolytically cleaved from their precursor molecule known as pro-opiomelanocortin (POMC) and post-translationally modified into melanocyte-stimulating hormones (α-, β- and γ-MSH) and adrenocorticotrophic hormone (ACTH) (Nakanishi et al, 1979; Cawley et al, 2016). These peptide hormones mediate their biological actions through five different but closely related G-protein coupled melanocortin receptors (MC1-R to MC5-R) that are distributed in many tissues and involved in the regulation of important physiological functions including skin pigmentation, steroidogenesis, energy homeostasis, sexual function, and exocrine secretion (Yang, 2011). Melanocortins have different affinities for their target receptors but α-MSH is unique in this

[1]Research Centre for Integrative Physiology & Pharmacology, Institute of Biomedicine, University of Turku, Turku, Finland. [2]Drug Research Doctoral Programme (DRDP), University of Turku, Turku, Finland. [3]Research Unit of Biomedicine and Internal Medicine, Department of Pharmacology and Toxicology, University of Oulu, Oulu, Finland. [4]Drug Research Program and Division of Pharmacology and Pharmacotherapy, Faculty of Pharmacy, University of Helsinki, Helsinki, Finland. [5]Institute for Molecular Medicine Finland (FIMM), HiLIFE Helsinki Institute of Life Science, University of Helsinki, Helsinki, Finland. [6]Faculty of Medicine, University of Turku, Turku, Finland. [7]Department of Chemistry and Biochemistry, University of Arizona, Tucson, AZ, USA. [8]Department of Medicine, Columbia University Vagelos College of Physicians and Surgeons, New York, NY, USA. [9]Turku Center for Disease Modeling, University of Turku, Turku, Finland. [10]Unit of Clinical Pharmacology, Turku University Hospital, Turku, Finland. [11]Medical Research Center Oulu, Oulu University Hospital and University of Oulu, Oulu, Finland. [12]Biocenter Oulu, University of Oulu, Oulu, Finland. ✉E-mail: pperin@utu.fi

regard as it has the ability to activate all MC-R subtypes except MC2-R (also known as ACTH receptor) (Yang, 2011; Schiöth et al, 1997). Classical physiological effects of α-MSH are skin pigmentation *via* MC1-R in melanocytes and regulation of energy homeostasis *via* MC4-R in the brain (Mountjoy et al, 1992; Cone, 2006). Owing to these actions, the principal sites of POMC expression and its processing into biologically active α-MSH lie within the central nervous system (hypothalamus and pituitary gland) and in the skin, but there is some evidence demonstrating that α-MSH is also produced in other peripheral tissues (Smith and Funder, 1988). Post-translational processing of POMC into mature α-MSH necessitates well-coordinated actions of several enzymes including carboxypeptidase E (CPE) and α-amidating monooxygenase (PAM), which co-localize in POMC-expressing cells (Cawley et al, 2016). It is generally thought that α-MSH is released into circulation by the pituitary gland, while in other tissues, α-MSH acts in an autocrine or paracrine fashion. Interestingly, early studies have discovered that *Pomc* mRNA and α-MSH production also occur in the rat heart (Millington et al, 1993, 1999). As a relevant clinical observation, plasma α-MSH level was found to be elevated in patients suffering from hypertrophic or dilated cardiomyopathy (Yamaoka-Tojo et al, 2006). However, the role of α-MSH and its possible target receptors in the heart are completely unknown.

Given the expression of α-MSH in the rat heart and increased plasma levels in heart failure patients, we hypothesized that its production in the heart is sensitive to pressure overload and modulated during the development of heart failure. We found that α-MSH production is significantly reduced in the failing mouse heart and that administration of a stable α-MSH analog protects the mice against pressure overload-induced cardiac hypertrophy and heart failure. Experimental data from a series of in vitro and in vivo experiments show that α-MSH acts as an anti-hypertrophic regulator by interacting with MC5-R in cardiac myocytes.

# Results

## α-MSH is expressed in the mouse heart and protects against pressure overload-induced pathological cardiac hypertrophy

We first aimed to investigate whether α-MSH is expressed in the mouse heart. Using an ELISA assay, we found detectable levels of α-MSH in different parts of the heart with higher expression levels appearing in the ventricles compared to the atria (Fig. 1A). Although α-MSH level was roughly 500-fold higher in the pituitary gland, which is the main source of α-MSH production, α-MSH concentration in other reference tissues such as the spleen and skeletal muscle was considerably lower compared to the heart (Fig. 1A). Cardiac α-MSH expression was also confirmed by immunohistochemistry, which revealed a small subset of cardiac cells that were positive for α-MSH (Fig. 1B). Using single-cell RNA-sequencing data from mice subjected to transverse aortic constriction (TAC) (Ren et al, 2020), we performed clustering analysis to identify *Pomc*-, *Cpe*-, and *Pam*-expressing cells in major cell types of the heart (Figs. 1C and EV1). Although *Pomc*+ cells were identified among all cell clusters, the majority of the triple-positive *Pomc*+ *Cpe*+ *Pam*+ cells, which are likely to produce POMC into mature α-MSH, were cardiomyocytes (Fig. 1D). As a technical note,

the number of *Pomc*+ and *Pomc*+ *Cpe*+ *Pam*+ cells was relatively low and probably underestimated due to the low detection sensitivity of scRNA-seq technology and consequent dropout of low-expression genes such as *Pomc*.

Next, to test the hypothesis that α-MSH production is triggered by pressure overload, we analyzed changes in cardiac *Pomc*, *Cpe*, and *Pam* expression at different stages of cardiac hypertrophy. The relative number of *Pomc*+ cells increased consistently in all major cell types during the early stage of cardiac hypertrophy (2 weeks after TAC) and then declined during the development of heart failure (5–8 weeks after TAC) (Fig. 1E). Likewise, in terms of changes in *Pomc*+ *Cpe*+ *Pam*+-cardiomyocyte count, there was a biphasic response with an initial elevation at 2 weeks post-TAC and then a decline in the failing heart (Fig. 1F). In good agreement with these findings, α-MSH concentration in the LV was reduced at the late stage of hypertrophy (5 weeks after TAC) (Fig. 1G), an effect occurring without a change in plasma α-MSH concentration (Fig. EV1). The TAC-induced reduction of ventricular α-MSH expression was also confirmed by Western blotting (Fig. EV1). In contrast, plasma or ventricular α-MSH level was not changed in mice subjected to angiotensin II (Ang II)-induced cardiac hypertrophy (Fig. EV1).

The declining level of α-MSH in the failing mouse heart raised a question whether α-MSH could be protective against pathological cardiac hypertrophy. To investigate the therapeutic potential of α-MSH, we subjected C57BL/6 J mice to TAC surgery and randomly assigned the mice to receive daily injections of either vehicle or a stable analog of α-MSH (melanotan-II; MT-II). After 8 weeks of TAC, α-MSH-treated mice showed attenuated LV hypertrophy compared to vehicle-treated TAC mice as evidenced by reduction in ventricular weight, ventricular weight-to-tibia length ratio (Fig. 1H–J) and ventricular weight-to-body weight ratio (Appendix Fig. S1). Histological examination also revealed reduced cardiomyocyte size and LV fibrosis in α-MSH-treated TAC mice (Fig. 1H,K,L). As determined by echocardiography, α-MSH treatment prevented TAC-induced thickening of LV posterior wall (Fig. 1M) and deterioration of LV ejection fraction (Fig. 1N). In sham-operated mice, α-MSH treatment induced a subtle but significant reduction in ventricular weight-to-tibia length ratio and LV posterior wall thickness without affecting LV ejection fraction (Fig. EV2). Lastly, gene expression analysis demonstrated that the molecular fingerprint of pathological cardiac hypertrophy was partly reversed by α-MSH with a significant reduction in TAC-induced expression of fibrosis-related genes including *Col1a2* (collagen type I, alpha 2) and *Mmp2* (matrix metalloproteinase-2) (Appendix Fig. S1). Taken together, these data indicate that local α-MSH production in the mouse heart is responsive to pressure overload and that α-MSH acts as an anti-hypertrophic regulator.

## α-MSH exerts an anti-hypertrophic effect in cultured cardiomyocytes

To investigate whether functional MC-Rs exist in cardiomyocytes, we performed experiments with H9c2 cells and neonatal mouse ventricular cardiac myocytes (NMCMs). Cells were treated with α-MSH for 5, 15, 30, or 60 min and assayed for intracellular cAMP levels since most MC-Rs are known to be coupled to G$_s$ proteins and cAMP signaling. Unexpectedly, α-MSH caused a reduction in cAMP level at 5 min time point in H9c2 cells and NMCMs

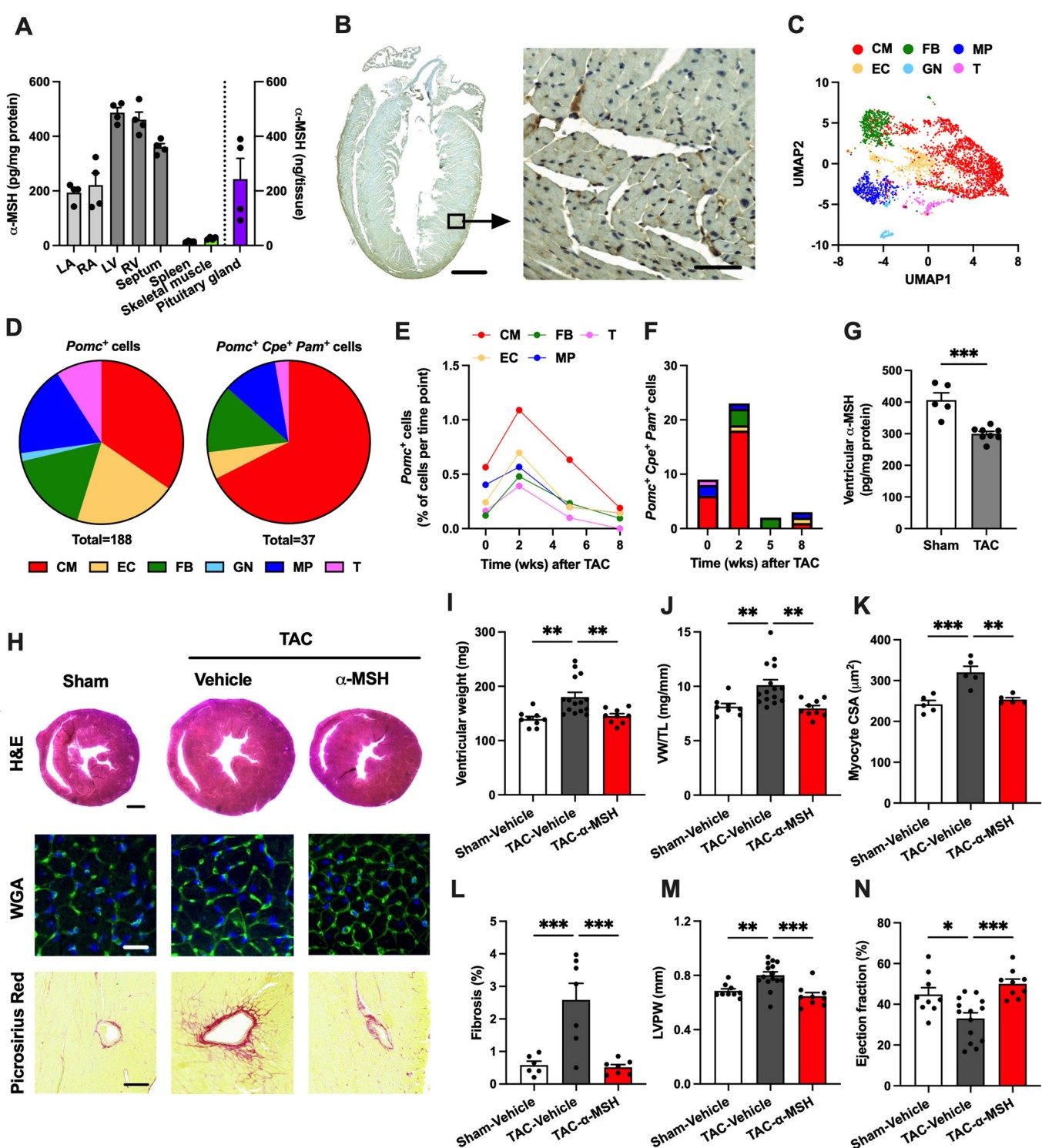

(Fig. 2A), indicating Gi-dependent coupling of the α-MSH response. Screening of other potential downstream targets of melanocortin signaling revealed a marked reduction in JNK phosphorylation after α-MSH treatment (Fig. 2B), while no effect was observed on phosphorylation of ERK1/2 (Fig. 2C) or intracellular Ca²⁺ responses (Appendix Fig. S2). In terms of concentration-responsiveness, α-MSH reduced intracellular cAMP level under baseline conditions (Fig. 2D) as well as in cells stimulated with the adenylyl cyclase activator forskolin (Fig. 2E) with effect peaking in the subnanomolar range of concentrations (<1 nM or log −9 M). JNK phosphorylation was also reduced at subnanomolar concentrations of α-MSH (Fig. 2F).

Finally, to investigate whether the activation of these signaling cascades leads to changes in gene expression, we performed qPCR

**Figure 1.  α-Melanocyte-stimulating hormone (α-MSH) is produced in the heart and protects against pathological cardiac hypertrophy.**

(A) α-MSH concentration (pg/mg protein) in the mouse left atrium (LA), right atrium (RA), left ventricle (LV), right ventricle (RV), septum, spleen, skeletal muscle, and pituitary gland (ng/whole gland, plotted on the right Y-axis). (B) Immunostaining of α-MSH in longitudinal heart section of C57Bl/6 J mouse. Scale bar, 1 mm (left) and 50 μm (right). (C) Uniform Manifold Approximation and Projection (UMAP) showing 11,492 single cells isolated from C57Bl mice at different stages of cardiac hypertrophy (0–8 weeks after TAC). Cell types were determined according to the expression of known markers. CM indicates cardiomyocyte; EC, endothelial cell; FB, fibroblast; GN, granulocyte; MP, macrophage; and T, T cell. (D) Pie charts showing the relative distribution of Pomc-positive (*Pomc*[+]) and triple-positive (*Pomc*[+] *Cpe*[+] *Pam*[+]) cells in each cell type. Cells were pooled from all time points (0–8 weeks after TAC). (E) Changes in the relative amount of *Pomc*[+] cells in each cell type as a function of time after transverse aortic constriction (TAC) surgery. *Pomc*[+] cells are expressed as percentage of total number of sequenced cells at each time point. (F) Changes in the number of *Pomc*[+] *Cpe*[+] *Pam*[+] cells in each cell type as a function of time after TAC surgery. (G) α-MSH concentration in the LV of sham- and TAC-operated mice 5 weeks after the surgery. \*\*\*$P < 0.001$ by Student's t test, $n = 5$ in sham and $n = 8$ in TAC. (H) Exemplary hematoxylin and eosin (H&E), wheat germ agglutinin (WGA), and Picrosirius Red-stained cross-sections of the heart of sham- and TAC-operated mice treated with either vehicle or α-MSH analog (melanotan II; MT-II). Scale bars, 1 mm (H&E), 25 μm (WGA) and 100 μm (Picrosirius Red). (I, J) Ventricular weight and ventricular weight to tibia length ratio (VW/TL) in the indicated groups. $n = 9$ in sham-vehicle, $n = 15$ in TAC-vehicle and $n = 9$ in TAC-α-MSH. (K) Quantification of cross-sectional area of ventricular cardiomyocytes. $n = 5$ in each group. (L) Quantification of the extent of LV fibrosis. $n = 6$ in sham-vehicle, $n = 7$ in TAC-vehicle and $n = 7$ in TAC-α-MSH. (M, N) Left ventricular posterior wall thickness (LVPW) and ejection fraction analyzed by echocardiography at the end of the experiment. $n = 9$ in sham-vehicle, $n = 15$ in TAC-vehicle and $n = 9$ in TAC-α-MSH. Data information: Data are mean ± SEM, each dot represents individual mouse. \*$P < 0.05$, \*\*$P < 0.01$, and \*\*\*$P < 0.001$ for the indicated comparisons by Student's t test (G) or by 1-way ANOVA and Dunnett post hoc tests (I–N). Source data are available online for this figure.

analysis for H9c2 cells and NMCMs that were treated with α-MSH for 1, 3, 6, or 24 h. We observed downregulation of the hypertrophy-related gene *Nppb* (B-type natriuretic peptide, Fig. 2G) and fibrosis-related genes including *Acta2* (alpha-smooth muscle actin, Fig. 2H), *Tgfb1* (transforming growth factor beta 1, Fig. 2I), *Col3a1* (collagen type III, alpha 1, Appendix Fig. S2) and *Fn1* (fibronectin, Appendix Fig. S2). In addition, the pro-inflammatory gene *Il6* (interleukin 6) was downregulated in NMCMs but not in H9c2 cells (Fig. 2J). In general, the changes in gene expression were short-lasting and vanished within 24 h, which probably reflects the short half-life of α-MSH.

To investigate the effects of α-MSH in hypertrophic context, we treated H9c2 cells with Ang II to promote cellular hypertrophy and used leucine incorporation assay as a measure of protein synthesis rate. α-MSH effectively prevented Ang II-induced increase in leucine incorporation (Fig. 2K). α-MSH also reversed the increase in JNK phosphorylation as well as in *Nppb* and *Acta2* expression in Ang II-treated cells (Fig. 2L–N). Collectively, the results demonstrate that functional MC-Rs exist in cardiomyocytes and suggest that the anti-hypertrophic effect observed in TAC-operated mice is dependent on direct actions of α-MSH on cardiomyocytes.

## MC5-R is expressed in mouse cardiomyocytes and downregulated during the development of heart failure

To investigate which MC-R subtype mediates the effects of α-MSH in cardiomyocytes, we used subtype selective MC-R agonists and screened their effects on gene expression by qPCR. We found that the MC5-R selective agonist PG-901 was the only compound that showed similar responses to α-MSH (Figs. 3A–D and EV3). Agonism at MC5-R downregulated *Tgfb1*, *Col3a1*, and *Fn1*, and tended to also reduce *Acta1* (actin, alpha skeletal muscle) and *Acta2* mRNA levels ($P = 0.08$ and 0.09, respectively) (Figs. 3A–D and EV3). We then sought to determine whether the α-MSH-induced effects are dependent on MC5-R activation in cardiomyocytes. Addition of the selective MC5-R antagonist PG-20N abolished the effect of α-MSH on p-JNK level and *Nppb* and *Col3a1* mRNA expression (Figs. 3E–H and EV3), further supporting the role of MC5-R as a mediator of the α-MSH-induced effects.

We next aimed to explore whether MC5-R is expressed in the mouse heart. Immunohistochemical staining revealed that MC5-R

is uniformly present in the heart (Fig. 3I). We also studied whether MC5-R protein abundance is changed in pressure-overloaded LV samples. For this purpose, we preselected samples from TAC-operated mice that displayed mild hypertrophy with normal LV systolic function (EF −3% vs sham, $P = 0.23$) and another set of mice which had more advanced hypertrophy and significant LV dysfunction (EF −25% vs sham, $P < 0.001$). Western blotting analysis revealed that in mildly hypertrophied LV samples (referred to as mTAC), MC5-R dimer, which is considered as functionally more active form, (Milligan, 2004) was significantly upregulated, while no change was found in the expression of MC5-R monomer (Fig. 3J,K). In contrast, MC5-R dimer was markedly reduced in severely hypertrophied LV samples (referred to as sTAC) and it was accompanied by a slight increase in the amount of MC5-R monomer (Fig. 3J,L). Cardiac hypertrophy induced by Ang II infusion (for 2 or 4 weeks) did not however affect MC5-R protein level in the LV (Fig. EV3).

## MC5-R acts as an anti-hypertrophic regulator in cultured cardiomyocytes

Since MC5-R appeared to be the target receptor for α-MSH in the heart, we turned our attention more closely to the MC5-R agonist PG-901 and investigated its actions on intracellular signaling cascades in cultured cardiomyocytes. In line with the effects evoked by α-MSH, PG-901 inhibited JNK pathway as evidenced by reduction in the phosphorylated form of JNK (Fig. 4A). No significant changes were observed in the phosphorylation of ERK1/2 (Fig. 4B) or intracellular $Ca^{2+}$ levels after PG-901 treatment (Appendix Fig. S3). In terms of JNK inhibition, the effect was most potent at 0.1 nM (log −10 M) concentration of PG-901 in H9c2 cells (Fig. 4C). PG-901 down-regulated fibrosis-associated genes such *Acta1*, *Fn1*, *Col3a1*, *Ctgf* (connective tissue growth factor) *Tgfb1* in H9c2 cells with a similar concentration-response profile to JNK inhibition (Appendix Fig. S3).

Mimicking the action of α-MSH, PG-901 also reduced intracellular cAMP level under baseline conditions (Fig. 4D) and to some extent also in forskolin-stimulated cells (Appendix Fig. S3). However, under a stronger and more physiological stimulus induced by the β-adrenoceptor agonist isoprenaline, PG-901 had no effect on cAMP level (Appendix Fig. S3), suggesting a rather weak coupling of MC5-R to Gi protein. Further mechanistic experiments using H9c2 cells revealed that inhibition of Gi

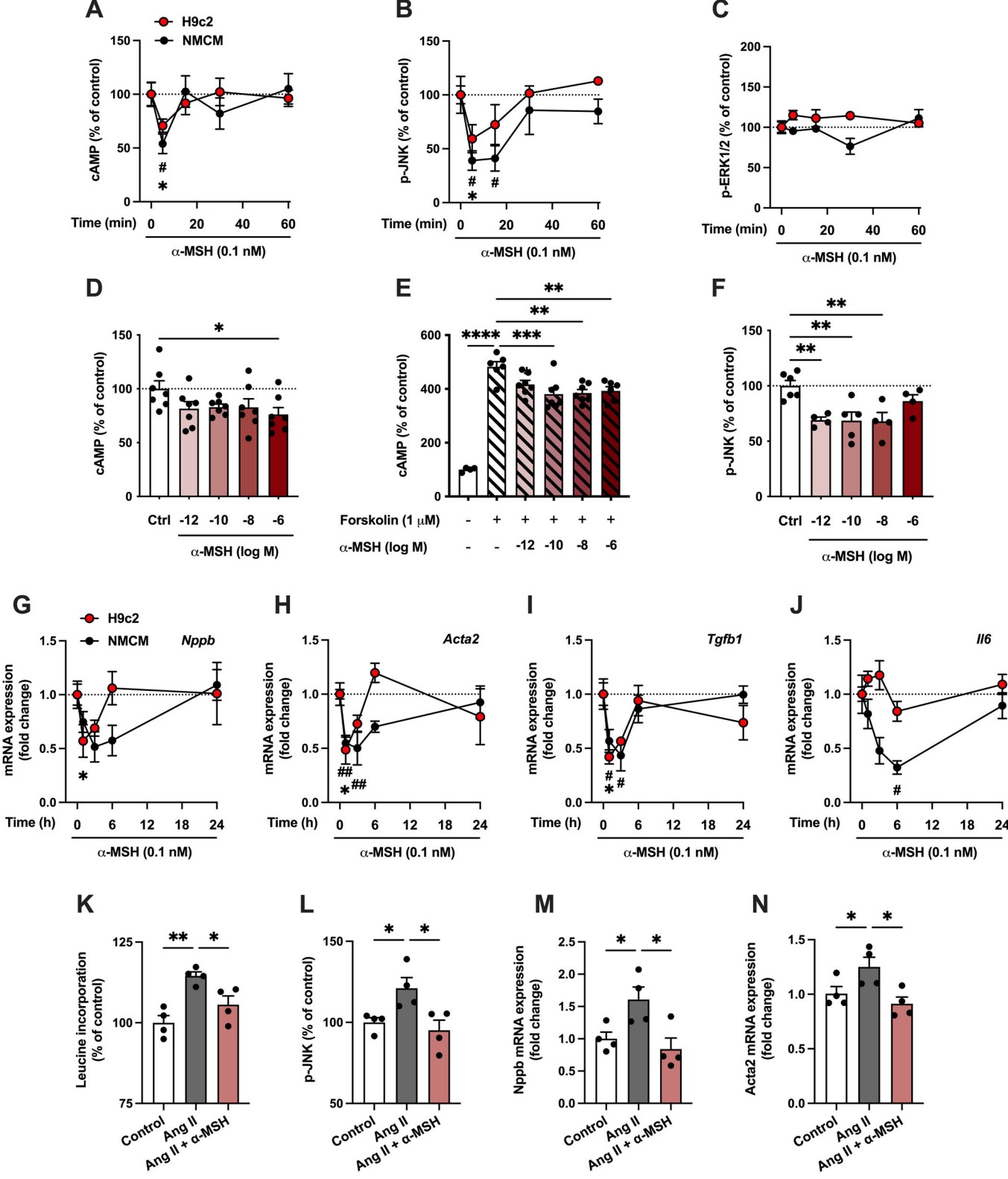

◄ **Figure 2. α-MSH reduces the levels of cAMP and phosphorylated JNK in cultured cardiomyocytes.**

(A–C) Quantification of intracellular cAMP levels in H9c2 cells and neonatal mouse ventricular cardiac myocytes (NMCMs) treated with α-MSH (0.1 nM) for 5, 15, 30, or 60 min (A). Data is expressed as percentage of control (Ctrl, 0 min). Quantification of phosphorylated JNK (B) and ERK1/2 (C) by ELISA assays in H9c2 cells and NMCMs treated with α-MSH (0.1 nM) for 5, 15, 30, or 60 min. *$P < 0.05$ versus Control (0 min) in H9c2 cells, #$P < 0.05$ versus Control (0 min) in NMCMs by 1-way ANOVA and Dunnett post hoc tests. (D–F) Quantification of intracellular cAMP levels in H9c2 cells treated with different concentrations of α-MSH for 30 min in the absence (D) or presence (E) of forskolin (1 μM). Quantification of phosphorylated JNK using ELISA assay in H9c2 cells treated with different concentrations of α-MSH for 30 min (F). *$P < 0.05$, **$P < 0.01$, ***$P < 0.001$, and ****$P < 0.0001$ for the indicated comparisons by 1-way ANOVA and Dunnett post hoc tests. (G–J) Quantitative real-time PCR (qPCR) analysis of *Nppb* (B-type natriuretic peptide), *Acta2* (alpha-smooth muscle actin), *Tgfb1* (transforming growth factor beta 1) and *Il6* (interleukin 6) in H9c2 cells and NMCMs treated with α-MSH (0.1 nM) for 1, 3, 6, or 24 h. *$P < 0.05$ versus Control (0 h) in H9c2 cells, #$P < 0.05$ and ##$P < 0.01$ versus Control (0 h) in NMCMs by 1-way ANOVA and Dunnett post hoc tests. (K–N) [³H]-Leucine incorporation assay (K) in H9c2 cells treated with Ang II (0.1 μM) for 24 h in the absence or presence of α-MSH (0.1 nM). Quantification of phosphorylated JNK by ELISA assay (L) and qPCR analysis of *Nppb* (L) and *Acta2* (M) expression in H9c2 cells treated with angiotensin II (Ang II, 0.1 μM) for 3 h in the absence or presence of α-MSH (0.1 nM). *$P < 0.05$ and **$P < 0.01$ for the indicated comparisons by 1-way ANOVA and Dunnett post hoc tests. Data information: Data are mean ± SEM, $n = 3$–7 per group (technical replicates) in each graph from 2–3 independent experiments. Source data are available online for this figure.

signaling with pertussis toxin (PTX) induced a further reduction in the amount of phosphorylated JNK and downregulation of *Nppb* and *Acta2*. Importantly, it did not abrogate the gene expression changes evoked by MC5-R activation (Appendix Fig. S4), indicating that signaling *via* Gi pathway does not mediate the downstream effects of MC5-R activation. Since MC5-R can simultaneously signal through both Gs- and Gi-dependent pathways, (Rodrigues et al, 2015) we also tested whether the effects of PG-901 could be reversed by blocking the cAMP-PKA axis. The PKA inhibitor H89 or a cAMP analog (cAMPS-Rp) that antagonizes cAMP-induced activation of PKA did not abolish the PG-901 evoked reduction of p-JNK or gene expression changes (Appendix Fig. S4). Taken together, these results demonstrate that the MC5-R-evoked changes in gene expression occur in a cAMP-independent manner.

To investigate whether MC5-R has a physiological significance in the regulation of cardiomyocyte growth, we used leucine incorporation assay and treated H9c2 cells with PG-901 in combination with different hypertrophic stimuli including Ang II, endothelin 1 (ET-1) and the α-adrenoceptor agonist phenylephrine (Phe). We found that PG-901 effectively prevented Ang II-induced increase in leucine incorporation (Fig. 4E), which is largely dependent on JNK activation (Yano et al, 1998). In contrast, no significant effect was observed on ET-1- or Phe-induced hypertrophic response (Fig. 4E), which has been shown to be more dependent on p38 phosphorylation (Nemoto et al, 1998). Likewise, in terms of gene expression, PG-901 blunted the induction of fibrotic genes such as *Tgfb1* and *Ctgf* most notably in Ang II-stimulated cells (Fig. 4F,G and Appendix Fig. S5). The inhibitory effect of PG-901 on leucine incorporation was also confirmed in Ang II-stimulated NMCMs (Fig. 4H).

We next investigated whether MC5-R silencing by siRNA causes an opposite phenotype to that seen after MC5-R activation. Indeed, MC5-R knockdown enhanced JNK phosphorylation (Fig. 4I) without significantly affecting other known intracellular signaling targets of MC-Rs (Fig. EV4) (Rodrigues et al, 2015; Xu et al, 2020). It was also associated with upregulation of *Nppa* (atrial natriuretic peptide), *Nppb*, *Acta1, Ctgf,* and *Fn1* (Fig. 4J) as well as with enhanced leucine incorporation (Fig. 4K). To test the dependency of the observed phenotype on JNK signaling, H9c2 cells were treated with the selective JNK inhibitor SP600125 prior to transfection with the *Mc5r* targeting siRNA. We observed that JNK inhibition with SP600125 completely abolished the increase in leucine incorporation and the upregulation of *Nppb* and *Ctgf* in *Mc5r*-silenced cells (Fig. 4L–N). The induction of *Nppb* expression

was similarly reversed by JNK inhibition in *Mc5r*-silenced NMCMs (Fig. EV4), further consolidating the link between MC5-R signaling, JNK pathway and cardiomyocyte hypertrophy. In addition, silencing of MC5-R signaling in NMCMs led to induction of pro-apoptotic genes such as *Casp3* (caspase-3), *Bax* (BCL2 associated X protein), and *Noxa* (phorbol-12-myristate-13-acetate-induced protein 1) as well as to enhanced protein expression of the pro-apoptotic p53 (Fig. EV4). However, these changes occurred in a JNK-independent manner (Fig. EV4).

## Human cardiomyocytes express functional MC5-R

The findings implicating an anti-hypertrophic role for MC5-R in mouse cardiomyocytes prompted us to investigate whether human cardiomyocytes also express MC5-R. We first quantified the mRNA levels of *MC5R* in the LV samples from control subjects and from patients with end-stage dilated (DCM) or ischemic cardiomyopathy (ICM). *MC5R* was significantly upregulated in the LV of DCM and ICM patients compared to controls (Fig. 5A). Likewise, *POMC* expression was increased in the diseased human hearts, particularly in the ICM patients (Fig. 5B). We also found that *MC5R* and *POMC* mRNA levels correlated positively and highly significantly in the DCM and ICM samples (Fig. 5C). We next studied whether *MC5R* and *POMC* expression are regulated by different hypertrophic stimuli in human induced pluripotent stem cell-derived cardiomyocytes (hiPSC-CM). First, hiPSC-CMs were exposed to mechanical stretch for 24 or 48 h (Pohjolainen et al, 2023), which led to a distinct gene expression pattern of the natriuretic peptides with a delayed upregulation of *NPPA* (after 48 h) and a more rapid induction of *NPPB* (after 24 h) (Appendix Fig. S6). Intriguingly, mechanical stretching of hiPSC-CMs downregulated *MC5R* expression but this occurred only after 48 h of stretch (Fig. 5D). In contrast, no change was observed in POMC expression (Fig. 5E). As another model of cardiomyocyte hypertrophy, hiPSC-CMs were treated with ET-1 (Pohjolainen et al, 2020), since Ang II or Phe does not evoke hypertrophic responses in these cells (Földes et al, 2014). Twenty-four-hour treatment with ET-1 led to stronger induction of *NPPA* and *NPPB* expression compared to mechanical stretching (Appendix Fig. S6). ET-1 treatment also clearly down-regulated *MC5R* (Fig. 5F), while *POMC* expression was unaffected (Fig. 5G), corroborating the finding from mechanical load-induced hypertrophy of hiPSC-CMs. Although no change was observed in *POMC* expression, *MC5R* expression positively correlated with *POMC* mRNA levels in the ET-1-treated hiPSC-CM samples

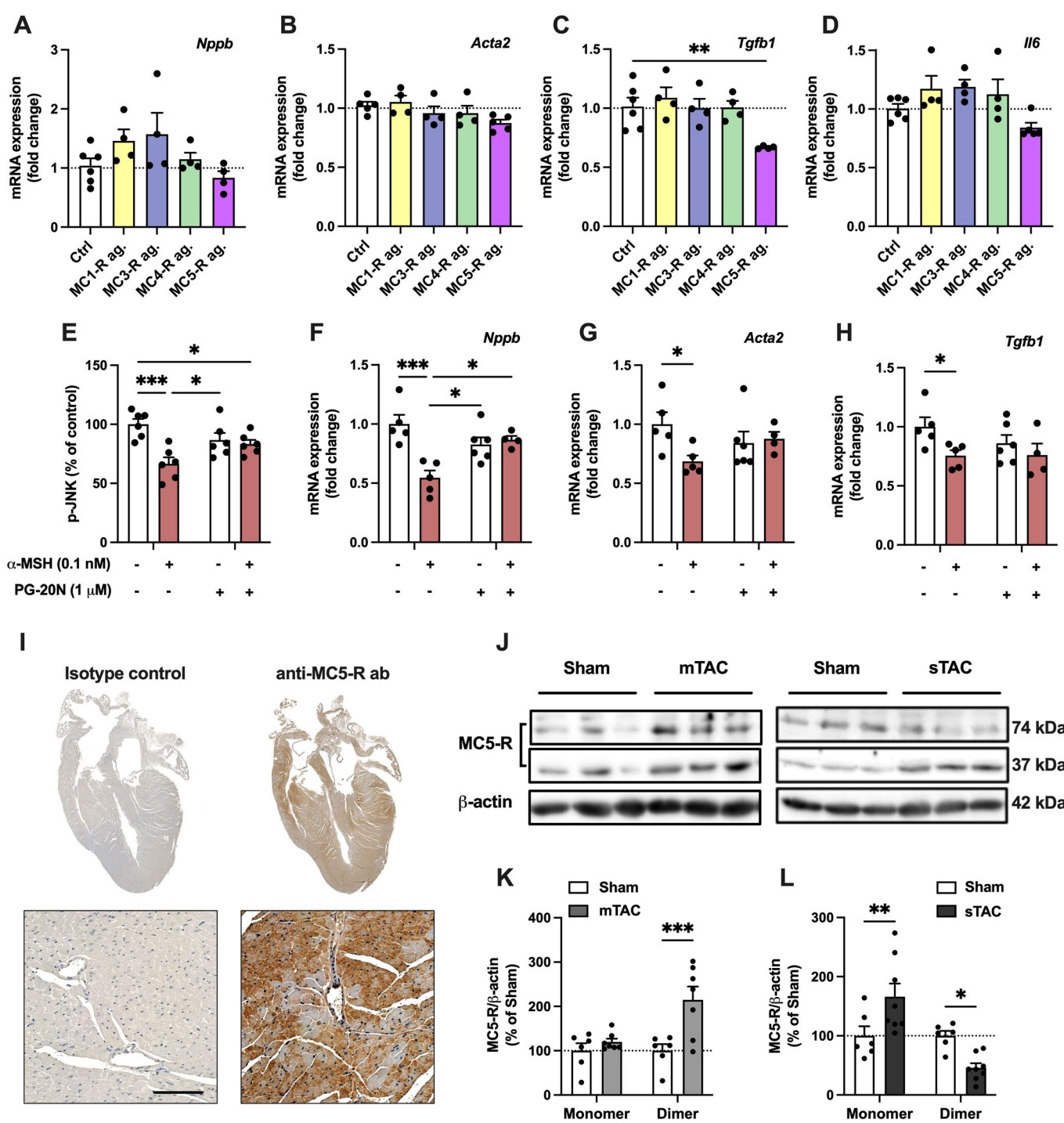

(Appendix Fig. S6), suggesting transcriptional co-regulation of these genes in cultured cardiomyocytes as well as in the human heart.

Since *MC5R* appeared to be expressed in hiPSC-CMs and regulated by hypertrophic stimuli, we next investigated the effects of the selective MC5-R agonist PG-901 in these cells under basal and ET-1-stimulated conditions. Quantification of proBNP protein expression, as a surrogate marker for cell hypertrophy, (Pohjolainen et al, 2020) revealed that PG-901 attenuated the hypertrophic effect of ET-1 in hiPSC-CMs (Fig. 5H). In terms of gene expression, PG-901 reduced *TGFB1, CTGF,* and *FN1* mRNA levels at 0.1 nM or 10 nM concentration in both unstimulated and stimulated hiPSC-CMs (Fig. 5I), while *ACTA1* was downregulated at the highest tested concentration (1 μM) in ET-1-stimulated cells (Fig. 5I). The expression of other genes of interest such as *NPPA* and *ACTA2* were unaffected by PG-901 treatment (Appendix Fig. S6). Taken together, these results demonstrate that MC5-R is functionally operative also in human cardiomyocytes.

◀ **Figure 3.  MC5-R is expressed in the mouse heart and its level is regulated by pressure overload.**

(A–D) Quantitative real-time PCR (qPCR) analysis of *Nppb* (B-type natriuretic peptide), *Acta2* (alpha-smooth muscle actin), *Tgfb1* (transforming growth factor beta 1) and *Il6* (interleukin 6) mRNA expression in H9c2 cells treated with subtype selective MC-R agonists for 3 h. $n = 4$–6 per group (technical replicates) in each graph from 2 independent experiments. (E–H) Quantification of phosphorylated JNK by ELISA assay and qPCR analysis of *Nppb*, *Acta2*, and *Tgfb1* expression in H9c2 cells treated with α-MSH (0.1 nM) for 1 h in the absence or presence of the selective MC5-R antagonist PG-20N (1 μM). $n = 4$–6 per group (technical replicates) in each graph from 2 independent experiments. (I) Immunostaining of MC5-R in longitudinal heart section of C57Bl/6 J mouse. In control section, anti-MC5-R antibody was replaced by purified normal rabbit IgG (isotype control). Scale bar, 100 μm. (J) Representative Western blots showing the monomer (37 kDa) and dimer (74 kDa) forms of MC5-R and β-actin (loading control) in the left ventricle (LV) after sham or TAC surgery. mTAC indicates mild TAC; sTAC, severe TAC. (K, L) Quantification of MC5-R monomer and dimer forms (normalized against β-actin) in mTAC and sTAC LV samples. $n = 6$–8 mice per group in each graph, each dot represents individual mouse. Data information: Data are mean ± SEM. *$P < 0.05$, **$P < 0.01$, ***$P < 0.001$ for the indicated comparisons by 1-way ANOVA and Dunnett post hoc tests (C) or by 2-way ANOVA and Bonferroni post hoc tests (E–H, K and L). Source data are available online for this figure.

## Cardiomyocyte-specific MC5-R deficiency aggravates pathological cardiac hypertrophy

To test for a regulatory role of MC5-R in cardiac hypertrophy, we engineered a tamoxifen-inducible cardiomyocyte-specific MC5-R KO mice (Mc5r-cKO) by crossing Mc5r^fl/fl mice with Myh6-MCM transgenic mice. Analysis of genomic DNA samples showed efficient Cre-lox recombination specifically in the heart after tamoxifen treatment that also resulted in ~50% reduction in cardiac MC5-R protein level (Appendix Fig. S7). In order to avoid Cre-mediated cardiotoxicity, tamoxifen dosing needs to be lowered and fractionated by administering tamoxifen on consecutive days, (Sohal et al, 2001; Hall et al, 2011) which could explain the partial recombination and gene knockdown in Mc5r-cKO.

Eight-week-old Mc5r-cKO mice and their age-matched controls (Mc5r^fl/fl and Myh6-MCM) were subjected to sham or TAC surgery and cardiac phenotyping was performed 4 weeks after surgery. No genotype difference was observed in sham-operated mice (Fig. 6A–D). However, Mc5r-cKO mice manifested a subtle but significant increase in the hypertrophic response to TAC surgery compared to control Mc5r^fl/fl and Myh6-MCM mice (Fig. 6A–D). Enhanced hypertrophic response was also apparent at the cellular level, where TAC-operated Mc5r-cKO mice demonstrated increased myocyte cross-sectional area (Fig. 6A,E). Supporting these findings, echocardiographic analysis revealed enhanced thickening of the LV posterior wall in Mc5r-cKO mice after TAC surgery (Fig. 6F). In terms of LV systolic function, TAC-operated mice did not show any deterioration of LV ejection fraction compared to sham-operated mice (Fig. 6G). However, tracing of the LV endocardial border to measure fractional area change (FAC) revealed depressed LV systolic function in TAC-operated mice and Mc5r-cKO mice showed lower FAC compared to Myh6-MCM mice among the TAC-operated groups (Fig. 6H). Furthermore, Mc5r-cKO mice showed reduced isovolumetric relaxation time (Fig. 6I) and increased mitral annular e'/a' ratio after TAC surgery (Appendix Fig. S8), while other parameters of diastolic function were unchanged (Appendix Fig. S8).

Corroborating the findings of MC5-R-mediated effects on fibrosis-associated genes in vitro, the extent of perivascular and interstitial fibrosis was increased in the LV of TAC-operated Mc5r-cKO mice (Fig. 7A,B). Gene expression analyses by qPCR also revealed upregulation of the hypertrophic marker genes *Nppa* and *Nppb*, fibrotic genes *Ctgf*, *Mmp2*, and *Fn1*, and the pro-inflammatory gene *Il6* in Mc5r-cKO mice after TAC surgery (Fig. 7C–H, Appendix Fig. S8). Furthermore, TAC-operated Mc5r-cKO showed enhanced number of apoptotic TUNEL-positive cells and increased protein expression of the pro-apoptotic BAX in the heart (Appendix Fig. S9).

In contrast to TAC model, 4-week infusion of Ang II induced a clear hypertrophic response but Mc5r-cKO mice were not sensitized to this response compared to their control genotypes (Appendix Fig. S10). Furthermore, cardiac function and structure, as assessed by echocardiography, did not reveal any genotype differences in Ang II-infused mice (Appendix Table S1).

## Pharmacological activation of MC5-R protects against heart failure

Finally, to evaluate the therapeutic potential of targeting MC5-R for the management of heart failure, we employed the TAC-model in C57Bl/6N mice that are more prone to develop TAC-induced heart failure (Garcia-Menendez et al, 2013; Zi et al, 2019) and treated the mice with PG-901 at two different dose levels (0.005 or 0.5 mg/kg/day). TAC-operated C57Bl/6N mice developed robust hypertrophic response in terms of ventricular weight (Fig. 8A–C), thickening of LV posterior wall (Fig. 8D) and LV dilatation (Appendix Fig. S11) but no significant treatment effect was noted for PG-901 in this regard. However, echocardiography revealed that TAC-operated mice treated with the low dose of PG-901 had a significant improvement in LV ejection fraction compared to vehicle-treated mice (Fig. 8E). PG-901 treatment also protected against TAC-induced reduction in mitral valve deceleration time (Fig. 8F), a measure of ventricular stiffness during diastole. Other parameters of diastolic function were unchanged in PG-901-treated mice compared to vehicle-treated TAC mice (Appendix Fig. S11). The observed functional changes were associated with a reduction in the extent of fibrosis (Fig. 8A,G) and number of apoptotic cells in the LV (Fig. EV5). In terms of molecular features of heart failure, low dose of PG-901 downregulated cardiac expression of *Nppa*, *Mmp2*, and *Ctgf* (Fig. 8H). In sham-operated mice, high or low dose of PG-901 had no significant effect on ventricular weight, LV structure or systolic function compared to vehicle-treated mice (Appendix Table S2). Taken together, these findings suggest that pharmacological targeting of MC5-R signaling could provide therapeutic benefits in the management of heart failure.

## Discussion

Our study uncovers a new melanocortin signaling pathway in the heart that is involved in the hypertrophic remodeling of the myocardium. First, we identified that *Pomc* and α-MSH expression in the heart is modulated by experimental pressure overload and showed that pharmacological treatment with α-MSH analog protects against pathological cardiac hypertrophy and LV systolic dysfunction. Secondly, our results establish a mechanistic link to

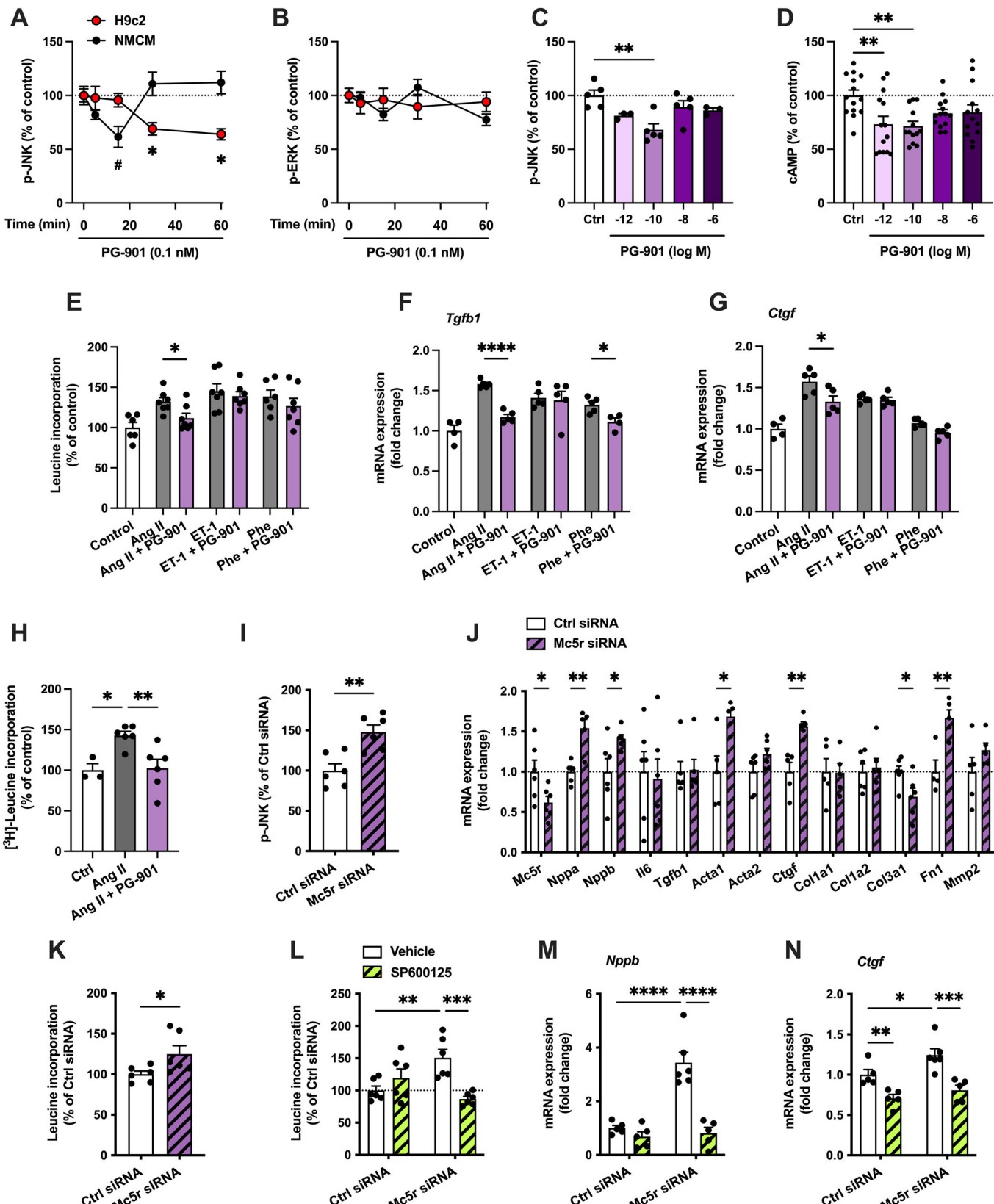

◀

**Figure 4. MC5-R activation with PG-901 mimics the actions of α-MSH in cultured cardiomyocytes.**

(A, B) Quantification of phosphorylated JNK and ERK by ELISA assays in H9c2 cells and NMCMs treated with PG-901 (0.1 nM) for 5, 15, 30, or 60 min. Data is expressed as percentage of control (0 min). * $P < 0.05$ versus Control (0 min) in H9c2 cells, #$P < 0.05$ versus Control (0 min) in NMCMs by 1-way ANOVA and Dunnett post hoc tests. $n = 4$-6 (technical replicates) per time point from 2 independent experiments. (C) Quantification of phosphorylated JNK using ELISA assay in H9c2 cells treated with different concentrations of PG-901 for 60 min. $n = 4$-6 (technical replicates) per group from 2 independent experiments. (D) Quantification of intracellular cAMP levels in H9c2 cells treated with different concentrations of PG-901 for 30 min. $n = 13$-14 (technical replicates) per group from 3 independent experiments. (E) [³H]-Leucine incorporation assay in H9c2 cells treated with angiotensin II (Ang II, 0.1 μM), endothelin 1 (ET-1, 0.1 μM) or phenylephrine (Phe, 0.1 mM) for 24 h in the absence or presence of PG-901 (0.1 nM). $n = 5$-7 (technical replicates) per group from 2 independent experiments. (F, G) Quantitative real-time PCR (qPCR) analysis of *Tgfb1* and *Ctgf* mRNA expression in H9c2 cells treated with Ang II, ET-1 or Phe for 3 h in the absence or presence of PG-901 (0.1 nM). $n = 4$-5 (technical replicates) per group from 2 independent experiments. (H) [³H]-Leucine incorporation assay in NMCMs treated with Ang II for 24 h in the absence or presence of PG-901 (0.1 nM). $n = 3$-6 (technical replicates) per group from 2 independent experiments. (I) Quantification of phosphorylated JNK using ELISA assay in H9c2 cells treated with control siRNA or *Mc5r* targeting siRNA for 24 h. $n = 6$ (technical replicates) per group from 2 independent experiments. (J) qPCR analysis of the indicated genes in NMCMs treated with control siRNA or *Mc5r* targeting siRNA for 24 h. $n = 4$-6 (technical replicates) per group from 2 independent experiments. (K) [³H]-Leucine incorporation in NMCMs treated with control siRNA or *Mc5r* targeting siRNA for 24 h. $n = 6$ (technical replicates) per group from 2 independent experiments. (L) [³H]-Leucine incorporation in H9c2 cells treated with or without the JNK inhibitor SP600125 (10 μM) for 30 min followed by transfection with control siRNA or *Mc5r* targeting siRNA for 24 h. $n = 6$ (technical replicates) per group from 2 independent experiments. (M, N) qPCR analysis of *Nppb* and *Ctgf* mRNA expression in H9c2 cells treated with or without the JNK inhibitor SP600125 (10 μM) for 30 min followed by transfection with control siRNA or *Mc5r* targeting siRNA for 24 h. $n = 6$ technical replicates) per group from 2 independent experiments. Data information: Data are mean ± SEM, *$P < 0.05$, **$P < 0.01$, ***$P < 0.001$ and ****$P < 0.0001$ for the indicated comparisons by Student's t-test (I-K), 1-way ANOVA and Dunnett post hoc tests (C-H and J) or 2-way ANOVA and Bonferroni post hoc tests (L-N). Source data are available online for this figure.

MC5-R that is functionally active in ventricular myocytes and mediates anti-hypertrophic and anti-fibrotic response upon activation. Conversely, silencing MC5-R signaling in cardiomyocytes aggravates pressure overload-induced cardiac hypertrophy and fibrosis. Taken together, these findings provide the first evidence that a functional melanocortin circuit exists in the heart and it regulates cardiac growth response in a favorable manner.

As a first line of evidence for the involvement of α-MSH in cardiac remodeling, we observed that acute pressure overload after TAC surgery triggered a change in cardiac *Pomc* and α-MSH expression with cardiomyocytes emerging as the primary source of α-MSH. It also appeared that there is a biphasic response in cardiac *Pomc* expression during the progression of heart failure with increased number of *Pomc*⁺ cells in compensated hypertrophy and then declining level in the failing heart. These results are in agreement with the clinical finding that plasma α-MSH level inversely correlated with NYHA functional class in heart failure patients (Yamaoka-Tojo et al, 2006). Considering that plasma α-MSH concentration is relatively low and it was not changed in TAC-operated mice, it is likely that α-MSH primarily acts in an autocrine or paracrine fashion in the heart without being significantly released into the circulation. To investigate whether the declining level of α-MSH in the failing heart could be counteracted by pharmacological means to provide therapeutic benefits, TAC-operated mice were chronically treated with α-MSH analog. Indeed, repeated α-MSH administration reduced ventricular weight and cardiac fibrosis and improved LV systolic function in TAC-challenged mice, demonstrating that α-MSH protects against pathological cardiac remodeling.

As a secreted peptide hormone, α-MSH may act on ventricular cardiac myocytes or other target cells in the heart such as fibroblasts, endothelial cells, or macrophages, which are all known to express functional MC-Rs. Although we cannot exclude the involvement of other cell types as mediators of the anti-hypertrophic and -fibrotic regulation of α-MSH, in vitro experiments with H9c2 cells and NMCMs proved that cardiomyocytes are responsive to α-MSH treatment and thus express functional MC-Rs. In the quest of the responsible MC-R subtype for the anti-hypertrophic regulation, we found that MC5-R activation mimicked the effects of α-MSH and that the effects of α-MSH

were reversed by MC5-R antagonism, supporting the notion that MC5-R is the primary receptor responsible for mediating the anti-hypertrophic regulation of α-MSH. Intriguingly, MC5-R is expressed in the mouse heart and the dimer form of MC5-R protein was increased in the LV of hypertrophied heart with normal ejection fraction, while in heart failure, the amount of MC5-R dimer form was significantly reduced. Thus, the progression from compensated hypertrophy to heart failure led to parallel changes in cardiac MC5-R expression and α-MSH level, suggesting that exhaustion of α-MSH production simultaneously compromises the integrity of MC5-R. Supporting this notion, a highly significant correlation between the expression of *MC5R* and *POMC* was observed in human LV samples and hiPSC-CMs. Of note, *MC5R* expression was upregulated in human DCM and ICM samples, while an opposite effect was observed in stretched and ET-1-stimulated hiPSC-CMs. There are multiple factors that might explain this discrepancy: e.g., age of subject, etiology of disease, chronic versus acute effect, and interfering signal from non-myocytes in the case of heart lysates.

Of particular importance, we found that functional MC5-R is expressed in mouse and human cardiomyocytes and identified a novel role for this MC-R subtype in cardiac remodeling. Previous studies have shown that MC5-R is most abundantly expressed in the skin, adrenal gland, and skeletal muscle, while low but detectable levels of *Mc5r* mRNA have been found in various peripheral tissues including the heart (Fathi et al, 1995; Barrett et al, 1994; Labbé et al, 1994). However, the functional significance of MC5-R in the heart has remained unexplored. A recent study that has the closest relevance to our work reported that H9c2 cells express MC5-R and respond to treatment with α-MSH or PG-901 at subnanomolar concentrations (Trotta et al, 2018). The authors aimed to build on the finding that α-MSH, by interacting with MC5-R, promotes glucose uptake in the skeletal muscle and expanded this concept to H9c2 cells by showing that MC5-R activation modulates the expression of glucose transporters and protects H9c2 cells against high glucose-induced apoptosis and hypertrophy (Trotta et al, 2018; Enriori et al, 2016). Our findings corroborate the existence of functional MC5-R in H9c2 cells and further prove that these cells respond to treatment with α-MSH or PG-901 in a similar way as primary cardiomyocytes. α-MSH and

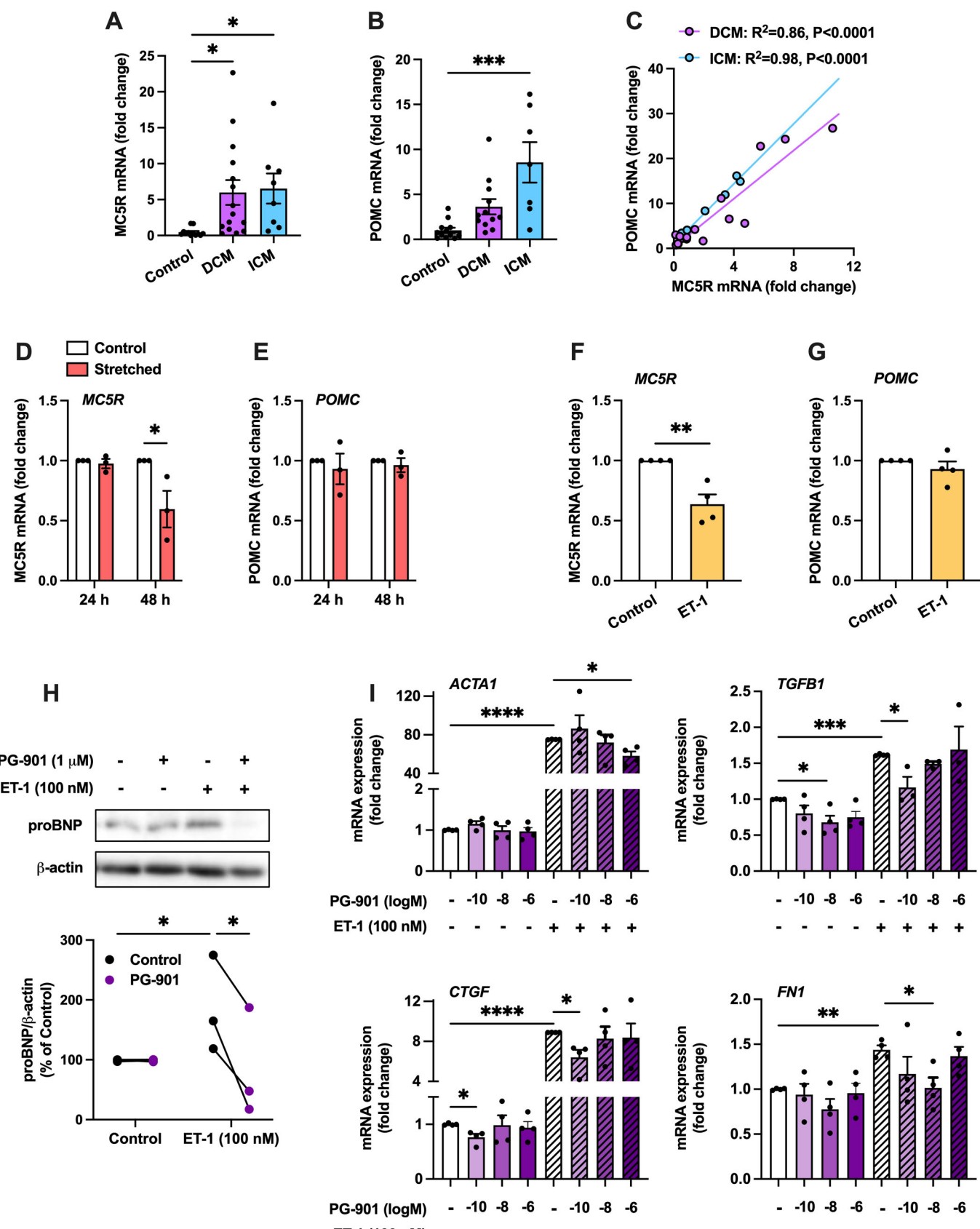

◄

**Figure 5. MC5R expression in human LV samples from cardiomyopathy patients and in human induced pluripotent stem cell-derived cardiomyocytes.**

(A, B) Quantitative PCR analysis of *MC5R* and *POMC* mRNA expression in left ventricular (LV) samples from healthy control subjects ($n = 13$) and from patients with end-stage dilated ($n = 14$) or ischemic ($n = 8$) cardiomyopathy. *$P < 0.05$ and ***$P < 0.001$ by one-way ANOVA and Dunnett post hoc tests. (C) Correlation between *MC5R* and *POMC* mRNA expression in DCM and ICM samples. Coefficients of determination ($R$ squared; $R^2$) and $P$ values by Pearson correlation are presented in the graph. (D, E) qPCR analysis of *MC5R* and *POMC* mRNA expression in human induced pluripotent stem cell-derived cardiomyocytes (hiPSC-CMs) that were mechanically stretched for 24 or 48 h. $n = 3$ individual experiments/batches of differentiation. *$P < 0.05$ versus Control by randomized block ANOVA (using individual experiments and treatment as factors). (F, G) qPCR analysis of *MC5R* and *POMC* mRNA expression in hiPSC-CMs treated with endothelin 1 (ET-1, 100 nM) for 24 h. $n = 4$ individual experiments/batches of differentiation. **$P < 0.01$ versus Control by randomized block ANOVA (using individual experiments and treatment as factors). (H) Representative Western blots and quantification of proBNP protein expression in hiPSC-CMs treated with PG-901 (1 μM) for 24 h in the absence or presence of ET-1 (100 nM). $n = 4$ individual experiments/batches of differentiation. Solid lines indicate trends within each individual experiment. *$P < 0.05$ for the indicated comparisons by randomized block ANOVA (using individual experiments and treatment as factors) and Dunnett post hoc tests. (I) qPCR analysis of *ACTA1*, *TGFB1*, *CTGF*, and *FN1* mRNA expression in hiPSC-CMs treated with different concentrations of PG-901 for 24 h in the absence or presence of ET-1 (100 nM). $n = 4$ individual experiments/batches of differentiation. *$P < 0.05$, **$P < 0.01$, ***$P < 0.001$, and ****$P < 0.0001$ for the indicated comparisons by randomized block ANOVA (using individual experiments and treatment as factors) and Dunnett post hoc tests. Data information: Data are mean ± SEM, each dot represents individual experiment. Source data are available online for this figure.

selective MC5-R agonism with PG-901 evoked similar responses in H9c2 cells and NMCMs but the effects of α-MSH appeared to be stronger compared to PG-901. Furthermore, the kinetic response profiles differed slightly with PG-901 showing more sustained effects in terms of JNK inhibition and downregulation of hypertrophy- and fibrosis-associated genes. This difference could be explained by the cyclic structure of PG-901 that makes it resistant to proteolytic degradation and thus biologically more stable compared to α-MSH (Grieco et al, 2002). PG-901 has been characterized to be a full agonist at the human MC5-R with an EC$_{50}$-value below 0.1 nM in a cAMP assay (Grieco et al, 2002), which is consistent with the present study showing appearance of effects at subnanomolar concentrations of PG-901. The in vitro results were also well in line with the dose-response relationship of PG-901 in TAC-operated mice. However, the lack of anti-hypertrophic effect of PG-901 in vivo is discordant with the phenotype of α-MSH-treated mice and the observed anti-hypertrophic regulation observed in vitro, which raises a concern that PG-901 might evoke an off-target effect. This, in turn, could mask the potential anti-hypertrophic effect of MC5-R activation in vivo. PG-901 is a full antagonist at the human MC3-R and MC4-R but its binding properties to mouse MC-R subtypes have not been reported, which leaves an open question whether PG-901 also interacts with other MC-Rs in the mouse heart.

Screening of the intracellular signaling responses revealed that MC5-R activation reduces cAMP level, indicative of Gi coupling. It has been previously established that MC5-R can engage two parallel signaling pathways upon activation: Gs/cAMP/PKA and Gi/ERK1/2 (Rodrigues et al, 2015). Despite the observed reduction in cAMP level, further mechanistic experiments showed that the downstream effects are independent of the Gi pathway. We also found that the signal transduction ensuing from MC5-R activation does not rely on Gs/cAMP/PKA axis either. Nevertheless, a more profound and consistent reduction was observed on the level of phosphorylated JNK after MC5-R activation. Intracellular signaling of MC-Rs has been rarely linked to the JNK pathway and its inhibition but a recent study demonstrated that melanocortin signaling through MC5-R can inhibit JNK activity in mouse adipocytes (Liu et al, 2017). The exact molecular mechanism for the JNK inhibition has remained elusive, but in the case of cardiomyocytes, it appears to be driven by a cAMP-independent signaling cascade.

JNKs belong to a subclass of stress-activated protein kinases (SAPKs) and in cardiomyocytes, their activation can be promoted by GPCRs, receptor tyrosine kinases and a variety of different stress

stimuli including oxidative stress and ischemia. Remarkably, in vitro and in vivo studies addressing the role of JNK signaling in cardiac hypertrophy have yielded conflicting results (Liang and Molkentin, 2003). Evidence obtained from in vitro studies strongly argue for a prohypertrophic role of JNKs, while loss-of-function approaches to silence JNK or its upstream regulators in vivo have resulted in promotion as well as attenuation of cardiac growth response (Minamino et al, 2002; Sadoshima et al, 2002; Liang et al, 2003). In vitro experiments in the current study demonstrate that MC5-R regulates hypertrophic growth of cardiomyocytes in a JNK-dependent manner. However, further studies are warranted to determine whether MC5-R-induced JNK reduction has a direct effect on cardiomyocyte growth in vivo.

Corroborating the regulatory role of MC5-R in cardiac remodeling, our loss-of-function study revealed that silencing MC5-R specifically in cardiomyocytes renders mice more susceptible to TAC-induced cardiac hypertrophy and fibrosis. As a limitation for drawing conclusion on the significance of MC5-R in cardiac remodeling, tamoxifen-induced Mc5r-cKO mice developed only a mild increase in ventricular weight after 4 weeks of TAC surgery compared to control mice. Due to insufficient recombination, Mc5r-cKO mice showed only a partial reduction (~50%) of MC5-R expression in the heart, which in turn, might explain the subtle phenotype of Mc5r-cKO mice and underestimate the role of MC5-R in cardiac remodeling. Another contributing factor might be that TAC-induced pressure overload was found to reduce the dimer form of MC5-R, thus limiting the incremental effect of genetically-induced MC5-R deficiency on the hypertrophic response.

Nevertheless, the cardiac phenotype of Mc5r-cKO mice matched closely the phenotype of cardiomyocytes with siRNA-induced knockdown of *Mc5r*. In vitro and in vivo silencing of MC5-R were both associated with upregulation of *Nppa*, *Nppb*, *Ctgf*, and *Fn1*. Conversely, MC5-R activation in vivo improved LV systolic function in TAC-operated mice and it was accompanied by reduced cardiac fibrosis and downregulation of *Nppa* and *Ctgf*. CTGF, for instance, is strongly produced by injured cardiomyocytes and it regulates many fibrosis-related processes such as extracellular matrix deposition (Chen et al, 2000; Dorn et al, 2018). CTGF also stimulates hypertrophic growth of cultured cardiomyocytes (Hayata et al, 2008; Yoon et al, 2010). Given the involvement of MC5-R in cardiac fibrosis, it will be intriguing to further explore whether functional MC5-R exists also in cardiac fibroblasts and could it synergize with MC5-R in cardiomyocytes to regulate remodeling of the myocardium. On the other hand, fibroblasts

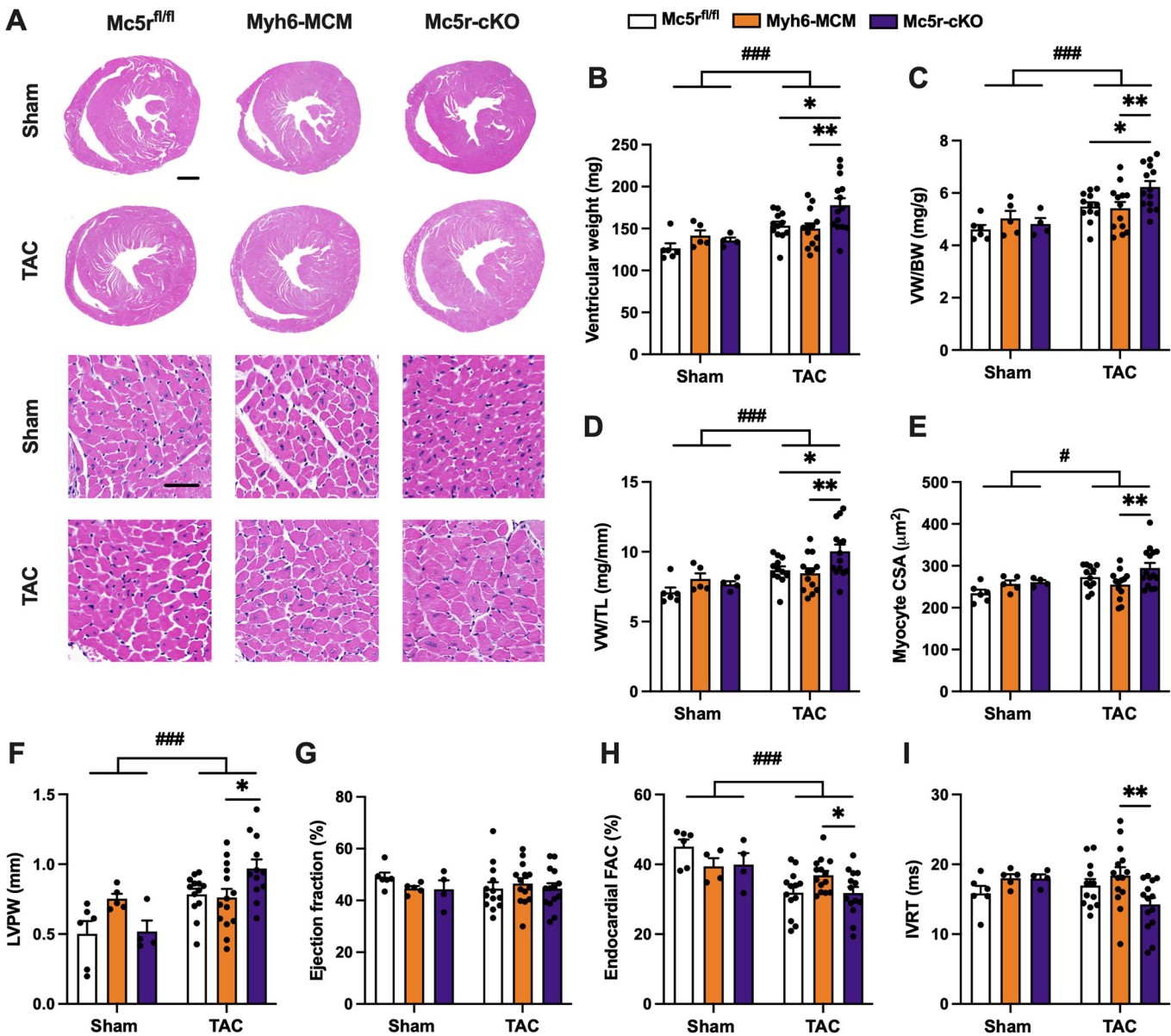

**Figure 6. Cardiomyocyte-restricted MC5-R deficiency aggravates cardiac hypertrophy after pressure overload.**

(A) Exemplary hematoxylin and eosin (H&E)-stained cross-sections of the heart showing the gross morphology of Mc5r^fl/fl, Myh6-MCM, and Mc5r-cKO mice after 4 weeks of sham or TAC operation. Scale bars, 1 mm (upper panel) and 20 μm (lower panel). (B–D) Ventricular weight, ventricular weight to tibia length ratio (VW/TL) and ventricular weight to body weight ratio (VW/BW) in the indicated groups. (E) Quantification of cross-sectional area of ventricular cardiomyocytes. (F–I) Echocardiographic analysis of LV posterior wall thickness (LVPW), ejection fraction, endocardial fractional area change (FAC) and isovolumetric relaxation time (IVRT) in Mc5r^fl/fl, Myh6-MCM, and Mc5r-cKO mice after 4 weeks of sham or TAC operation. Data information: Data are mean ± SEM, each dot represents individual mouse. $n = 6$ in sham Mc5r^fl/fl mice, $n = 4$–5 in sham Myh6-MCM mice, $n = 4$ in sham Mc5r-cKO mice, $n = 12$ in TAC Mc5r^fl/fl mice, $n = 13$ in TAC Myh6-MCM mice and $n = 14$ in TAC Mc5r^fl/fl mice. *$P < 0.05$ and **$P < 0.01$ for the indicated comparisons by 2-way ANOVA and Dunnett post hoc tests. #$P < 0.05$, ##$P < 0.01$, ###$P < 0.001$ for the main effect of TAC by 2-way ANOVA. Source data are available online for this figure.

are considered as important effector cells in the hypertrophic response to Ang II and Mc5r-cKO mice were not sensitized to Ang II-induced cardiac hypertrophy (Booz and Baker, 1995; Bouzegrhane and Thibault, 2002). Furthermore, Ang II infusion did not change MC5-R expression in the heart, while TAC surgery clearly affected cardiac MC5-R protein levels. These findings suggest that MC5-R signaling does not modulate hypertrophic remodeling that is primarily driven by cardiac fibroblasts or other non-myocytes.

In conclusion, the present study uncovers a novel role for α-MSH and MC5-R in pathological cardiac remodeling. α-MSH is expressed in the heart and protects against pathological cardiac hypertrophy by activating MC5-R in cardiac myocytes, which may be a potential therapeutic target for the management of heart failure. Considering that analogs of naturally occurring α-MSH have been recently approved for clinical use (Scenesse®, Vyleesi®, and Imcivree®), (Montero-Melendez et al, 2022) it is important to

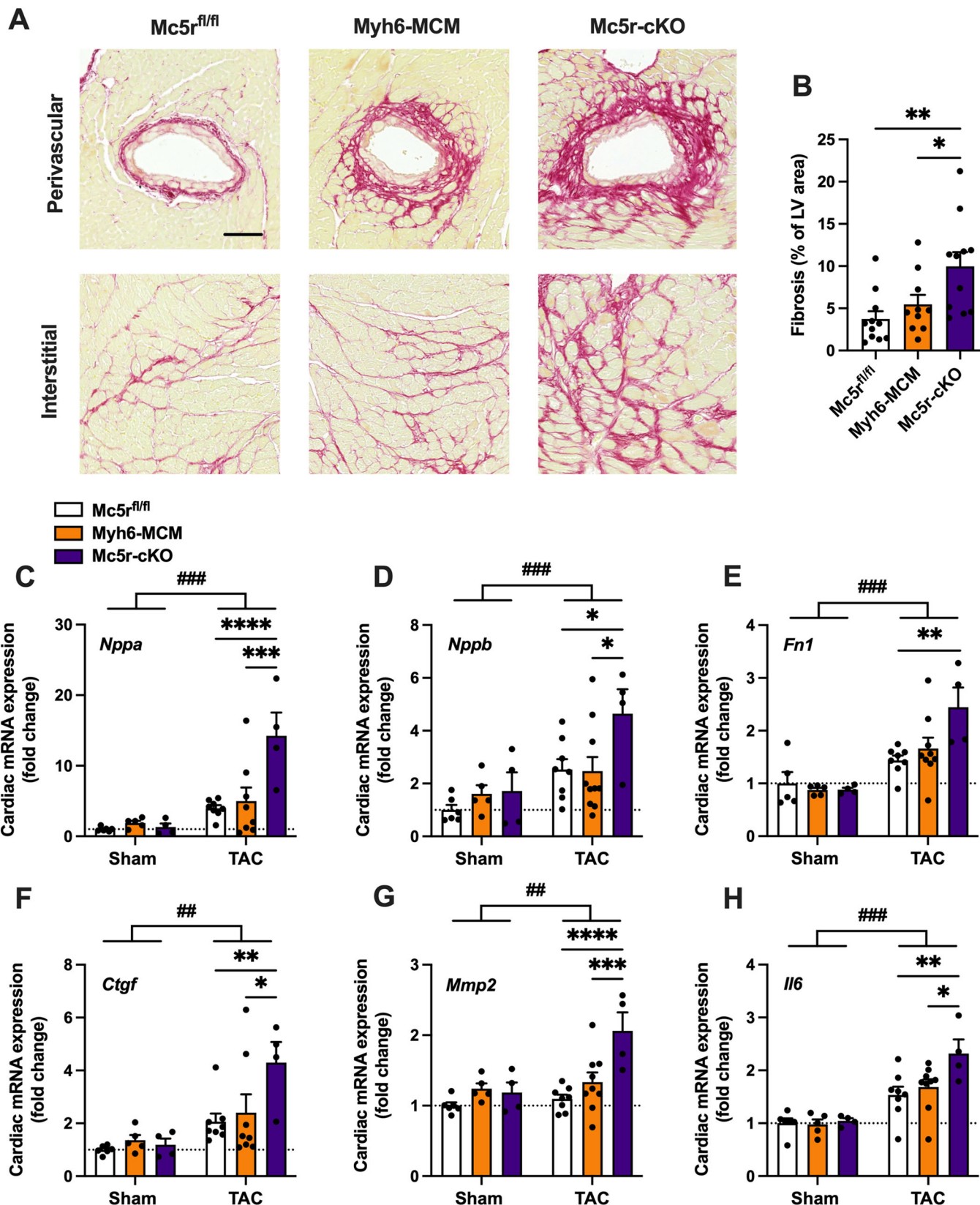

◄ **Figure 7. Cardiomyocyte-restricted MC5-R knockout mice show enhanced cardiac fibrosis after TAC surgery.**

(A) Exemplary Picrosirius Red-stained cross-sections of the LV showing the extent of perivascular and interstitial fibrosis in Mc5r[fl/fl], Myh6-MCM, and Mc5r-cKO mice after TAC surgery. Scale bar, 100 µm. (B) Comparison of the LV collagen area in Mc5r[fl/fl], Myh6-MCM, and Mc5r-cKO mice after TAC surgery. $n = 11$ in Mc5r[fl/fl] mice, $n = 10$ in Myh6-MCM mice and $n = 11$ in Mc5r[fl/fl] mice. (C–H) Quantitative real-time PCR (qPCR) analysis of *Nppa* (atrial natriuretic peptide), *Nppb* (brain B-type peptide), *Fn1* (fibronectin), *Ctgf* (connective tissue growth factor), *Mmp2* (matrix metalloproteinase 2) and *Il6* (interleukin 6) in the LV of Mc5r[fl/fl], Myh6-MCM and Mc5r-cKO mice after sham or TAC surgery. $n = 6$ in sham Mc5r[fl/fl] mice, $n = 5$ in sham Myh6-MCM mice, $n = 4$ in sham Mc5r-cKO mice, $n = 8$ in TAC Mc5r[fl/fl] mice, $n = 9$ in TAC Myh6-MCM mice and $n = 4$ in TAC Mc5r[fl/fl] mice. Data information: Data are mean ± SEM, each dot represents individual mouse. $*P < 0.05$, $**P < 0.01$, $***P < 0.001$, and $****P < 0.0001$ for the indicated comparisons by 2-way ANOVA and Dunnet post hoc tests. $\#\#P < 0.01$, $\#\#\#P < 0.001$ for the main effect of TAC by 2-way ANOVA. Source data are available online for this figure.

further evaluate the cardiovascular safety of melanocortin drugs. Even if future research does not support the translation of MC5-R targeted drugs for treating human heart failure, the present findings predict a favorable cardiac safety profile for drugs that have agonistic activity at MC5-R.

# Methods

## Mice and study design

Mice were housed in groups on a 12 h light/dark cycle with free access to food (# 2916C, Teklad Global diet, Envigo) and tap water. The experiments were approved by the national Animal Experiment Board in Finland (License number: ESAVI/6280/04.10.07/2016 and ESAVI/1260/2020) and conducted in accordance with the Directive 2010/63/EU of the European Parliament on the protection of animals used for scientific purposes and with the institutional and national guidelines for the care and use of laboratory animals. Sample sizes were empirically determined based on previous experience with the experimental models. In pharmacological in vivo experiments, mice were allocated to the treatment groups using simple randomization. Where possible, experiments were conducted and analyzed by blinded researchers.

For non-selective activation of the melanocortin system in vivo, 8-week-old male C57BL/6J mice were subjected to transverse aortic constriction (TAC) as described below. Mice were allowed to recover for 2 weeks from the surgery and were thereafter randomly assigned to receive daily i.p. injections of either vehicle (PBS) or the α-MSH analog melanotan-II (0.3 mg/kg/day, Tocris, # 2566) (Al-Obeidi et al, 1989). The selected dose was previously shown to be safe and therapeutically effective in other experimental models of disease such as atherosclerosis (Rinne et al, 2013, 2014). Sham-operated mice received vehicle or melanotan-II according to the same treatment scheme. Mice were sacrificed 8 weeks after the TAC operation.

To generate inducible cardiomyocyte-specific MC5-R knockout (Mc5r-cKO) mice, MC5-R floxed mice (Mc5r[fl/fl], GemPharmatec, strain # T00591) were intercrossed with tamoxifen-inducible Myh6-MerCreMer transgenic mice (Myh6-MCM, the Jackson Laboratory, strain # 005657) (Sohal et al, 2001). All mice were on C57Bl/6 J background. At 6 weeks of age (range 5–7 weeks), male mice were treated with tamoxifen (20 mg/kg, i.p., Cayman Chemicals, Ann Arbor, MI, # 13258) on 4 consecutive days to induce Cre-mediated recombination. Tamoxifen was dissolved in peanut oil by heating at +37 °C. Age-matched Mc5r[fl/fl] (Myh6-MCM[-/-]) and Myh6-MCM (Mc5r[wt/wt]) mice were used as controls and treated with tamoxifen as described above. Mice were allowed

to recover at least for 7 days from tamoxifen treatment before any experimentation. At the end of the experiment, genomic DNA samples from the heart and skeletal muscle (control tissue) were genotyped for the recombined allele using the following primers: 5'-ATT GAG GAT TCG AGG AGA GTC CTG-3' and 5'-AAG CCA TAG GGC CAG AAG TCT AG-3'. The size of the PCR product for the recombined allele was ~300 kB.

To study the therapeutic benefits of MC5-R activation in a heart failure model, 8-week-old male C57BL/6N were subjected to TAC and after a one-week recovery period, randomly assigned to receive i.p. injections of either vehicle (PBS) or the selective MC5-R agonist PG-901 (0.005 or 0.5 mg/kg/day). Mice were sacrificed 5 weeks after the TAC operation. At the end of the experiments, mice were euthanized *via* $CO_2$ asphyxiation and whole blood was obtained *via* cardiac puncture. Heart was weighed and collected for further analyses.

## Cardiac hypertrophy models and echocardiography

To induce hemodynamic pressure overload, mice were subjected to TAC as previously described (Szabó et al, 2014). Briefly, mice were anesthetized, intubated, and ventilated (MiniVent, Harvard Apparatus) for the surgery. Median sternotomy was performed, the transverse aorta was revealed and ligated with a 27-G needle and a 7-0 silk suture. The thoracic cage and skin were closed with 6-0 surgical silk sutures. Sham-operated mice underwent the same procedure without constriction of the aorta and served as controls. As another model of cardiac hypertrophy, mice were subjected to subcutaneous infusion of angiotensin II (1.4 mg/kg/day) for 2 or 4 weeks using osmotic minipumps (Alzet, Model 1004) (Szabó et al, 2014). For the surgical operations, mice were anesthetized with ketamine (110 mg/kg, i.p.) and xylazine (15 mg/kg, i.p.), and buprenorphine (0.05 mg/kg, s.c., 2x/day for 3 days) and carprofen (5 mg/kg, s.c., 1x/day for 3 days) were given for peri- and post-operative analgesia.

Cardiac structure and function were assessed by transthoracic echocardiography (Vevo 2100, Visual Sonics Inc., Toronto, Canada) before the start of drug administration and at the end of the experiment under isoflurane anesthesia (4% for induction and 2% for maintenance). B-mode, M-mode, transmitral pulsed wave and tissue Doppler images were recorded and analyzed with Vevo software (Vevo LAB 5.5.0) by a blinded observer. Echocardiography measurements of cardiac structure and function prior to drug administration are reported in Appendix Tables S3 and S4.

## Measurement of tissue α-MSH levels

Plasma, atria, and ventricles of the heart were harvested from sham- and TAC-operated mice and assayed for α-MSH using a

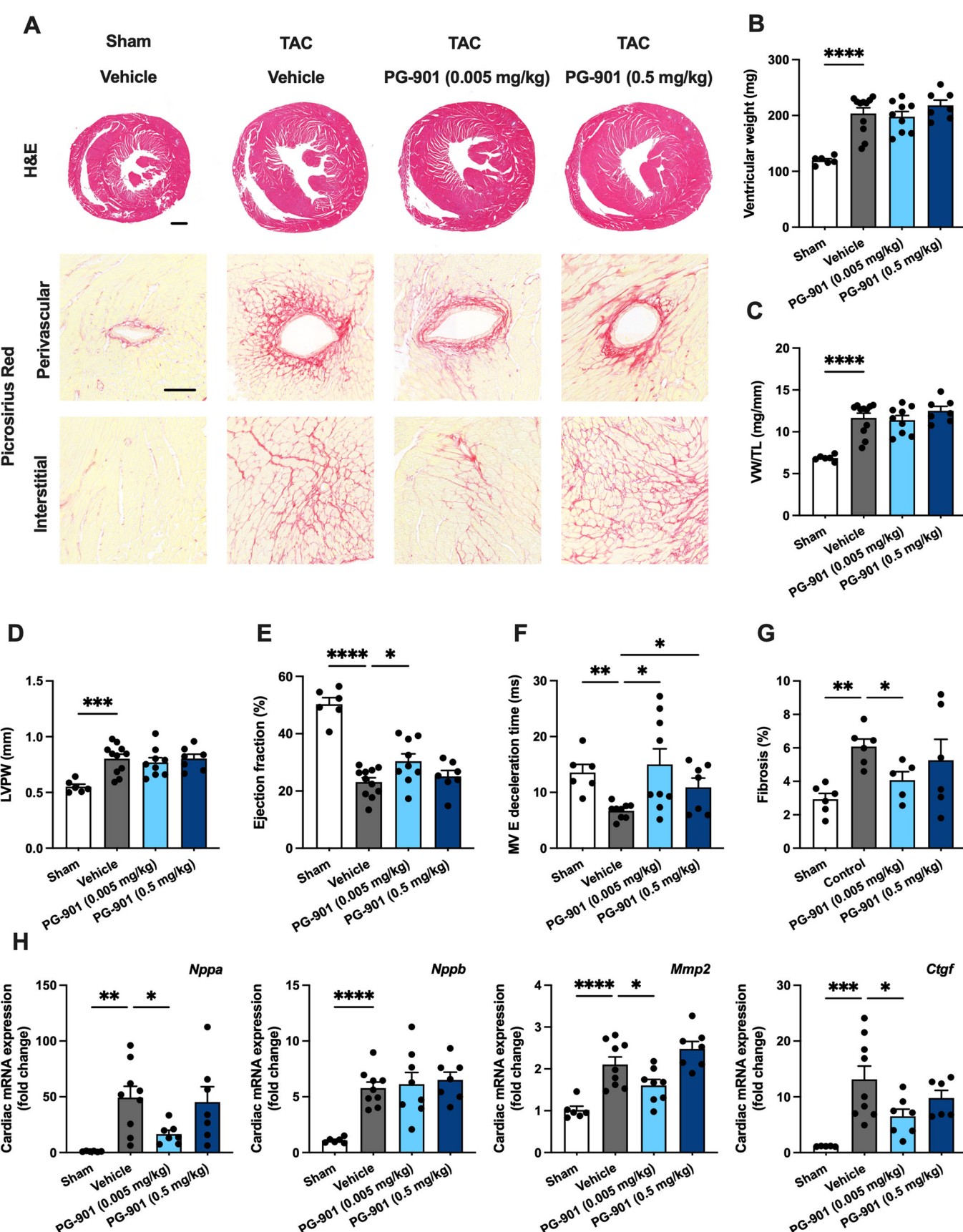

**Figure 8.   MC5-R activation improves LV systolic function and reduces cardiac fibrosis in TAC-operated mice.**

(A) Exemplary hematoxylin and eosin (H&E)- and Picrosirius Red-stained cross-sections of the heart showing the gross morphology and the extent of perivascular and interstitial fibrosis in sham- and TAC-operated mice treated with either vehicle or PG-901 (0.5 or 0.005 mg/kg/day). Scale bar, 1 mm (H&E), 100 µm (Picrosirius Red). (B, C) Ventricular weight and ventricular weight to tibia length ratio (VW/TL) in the indicated groups. (D, E) Left ventricular posterior wall thickness (LVPW) and ejection fraction analyzed by echocardiography at the end of the experiment. (F) Mitral valve (MV) deceleration time analyzed by pulsed-wave Doppler echocardiography at the end of the experiment. (G) Quantification of the extent of LV fibrosis. (H) Quantitative real-time PCR (qPCR) analysis of *Nppa* (atrial natriuretic peptide), *Nppb* (B-type natriuretic peptide), *Mmp2* (matrix metalloproteinase 2), and *Ctgf* (connective tissue growth factor) in the LV of sham- and TAC-operated mice treated with either vehicle or PG-901. Data information: Data are mean ± SEM, each dot represents individual mouse. (B–F, H): $n = 6$ in Sham, $n = 9$–11 in TAC/Vehicle, $n = 7$–9 in TAC/PG-901 (0.005 mg/kg), and $n = 6$–7 in TAC/PG-901 (0.5 mg/kg). (G): $n = 5$–6 mice per group. *$P < 0.05$, **$P < 0.01$, ***$P < 0.001$, and ****$P < 0.0001$ for the indicated comparisons by 1-way ANOVA and Dunnett post hoc tests. Source data are available online for this figure.

commercial ELISA assay (antibodies-online GmbH, #ABIN6969629). Tissue samples were homogenized in PBS supplemented with a protease inhibitor cocktail (Complete Mini, Roche) and centrifuged for 5 min at $5000 \times g$. After centrifugation, the resultant supernatant was diluted with assay buffer and assayed for α-MSH concentration. The results were normalized against total protein concentrations (Pierce™ BCA Protein Assay Kit, ThermoFisher).

## Single-cell RNA-sequencing analysis

Single-cell RNA-sequencing data deposited in the Gene Expression Omnibus (GSE120064) was used to study *Pomc-*, *Cpe-* and *Pam-* expressing cell types in the mouse heart. All quality control passed cells (11,492) were analyzed using Seurat (version 4.0.3) with R (version 4.2.0) (Hao et al, 2021). Raw gene counts were first normalized using LogNormalize method and then scaled using a scale factor of 10,000. An unsupervised dimensional reduction method, UMAP (Uniform Manifold Approximation and Projection) was used to generate clustering results using first 26 principal components. The same cell type annotation as reported by the original authors was used in the analysis (Ren et al, 2020).

## Cell culture and treatments

Rat heart myoblast H9c2(2-1) cells (ATCC®, CRL-1446™) were cultured in Dulbecco's Modified Eagle's Medium (DMEM; Merck Life Science Oy) supplemented with 1% penicillin-streptomycin (Gibco, USA, #15140-122) and 10% heat-inactivated fetal bovine serum (Biowest, South America Origin, # S181B-500 or Gibco, UK, # 10270-106) at $+37\,°C$ under 5% $CO_2$ level. Cells were subcultured with a ratio of 1:3 to 1:5 when confluency reached 60–70%.

Neonatal mouse ventricular cardiac myocytes (NMCM) were isolated from 1- to 3-day-old C57BL/6NCrl mouse pups using the Pierce Primary Cardiomyocyte Isolation Kit (ThermoFisher, # 88281). Cells were cultured according to the manufacturer's protocol and used in experiments after 5 to 7 days of culturing.

Human induced pluripotent stem cell-derived cardiomyocytes (hiPSC-CM) were produced from iPS (IMR90)-4 line (WiCell, Madison, Wisconsin, United States) as previously described (Burridge et al, 2014; Karhu et al, 2018). Briefly, the hiPSCs were cultured in Essential 8 medium (Gibco) and passaged approximately every 4 days using Versene (Gibco). Differentiation was carried out in RPMI1640 medium (Gibco) supplemented with B-27 minus insulin (Gibco). Differentiation was induced at a confluence of 80–90% (day 0) with a 24-h exposure to 6 µM CHIR99021 (Tocris Bioscience), and on day 3 the mesodermal progenitors were

directed towards cardiac lineage with a 48-h exposure to 2.5 µM Wnt-C59, whereafter medium was replenished every 2 days. From day 11 to day 15 the differentiated cells were deprived of glucose by culturing them in RPMI1640 without glucose (Gibco) supplemented with B-27 supplement (containing insulin; Gibco) to purify the culture of non-myocytes. On day 15, the purity of hiPSC-CM cultures was assessed visually based on beating and only differentiations with a purity of >95% were used for the experiments. For drug treatments and qPCR analyses, hiPSC-CM were seeded on gelatin- or Matrigel-coated 12-well plates at 500,000 cells/well in medium containing RPMI 1640, B-27 supplement, 10 µM ROCK inhibitor and 10% fetal bovine serum. The cells were let to attach for 48 h and were thereafter maintained in RPMI 1640 medium supplemented with B-27 until 30-52 days old from the beginning of differentiation before commencing the experiments.

H9c2 cells, NMCMs, and hiPSC-CM were used to study the effects of α-MSH (Abcam, # ab120189) and selective MC-R agonists in vitro. LD211 was used as a selective agonist for MC1-R (compound 28 in the original publication) (Doedens et al, 2010), [D-Trp8]-γ-MSH as a selective MC3-R agonist (Grieco et al, 2000), THIQ as a selective MC4-R agonist (Cayman Chemical, USA, # 312637-48-2) (Sebhat et al, 2002), and PG-901 as a selective MC5-R agonist (Grieco et al, 2002). PG-20N was used as a selective MC5-R antagonist (Grieco et al, 2008). LD211, [D-Trp8]-γ-MSH, PG-901, and PG-20N were synthetized and provided by Professor Minying Cai.

To induce cell hypertrophy, cells were treated with angiotensin II (Ang II, Abcam, # ab120183), endothelin 1 (ET-1, Merck Life Science Oy, # E7764) or phenylephrine (Phe, Merck Life Science Oy, # P6126) for 24 h. As another model of cell hypertrophy, hiPSC-CMs were seeded on Matrigel-coated 6-well Bioflex® plates at 750,000–1,000,000 cells/well and exposed to cyclic mechanical stretch by applying vacuum suction with an FX-5000 Tension System (Flexcell International Corporation, Hillsborough, NC, USA). Cells were stretched for 24 h or 48 h in two-second cycles (0.5 Hz) of sinusoidal wave varying between 10–21% equibiaxial stretch. Control samples were from the same differentiation and simultaneously cultured on Matrigel-coated Bioflex® plates without applying stretch.

For siRNA-mediated knockdown of MC5-R, NMCMs and H9c2 cells were transfected using Lipofectamine RNAiMAX (Thermo-Fisher, # 13778030) and MC5-R siRNA (ThermoFisher, Silencer™ Select Mc5r siRNA, # 4390771, assay ID s69672 for NMCM or s130589 & s130590 for H9c2) or negative control siRNA (Silencer™ Select Negative Control No. 1 siRNA, #4390843) that was diluted in Opti-MEM medium (ThermoFisher, # 31985062). Cells were

cultured with siRNA-lipid complexes 24 to 48 h before collection and analyses. To inhibit the JNK pathway, H9c2 cells, and NMCMs were treated with SP 600125 (10 μM, Tocris, #1496) for 30 min before applying other treatments as indicated in the figure legends.

## Human cardiomyopathy samples

Human LV samples were obtained from dilated ($n = 15$) and ischemic ($n = 8$) cardiomyopathy patients undergoing cardiac transplantation in Helsinki University Hospital between 2014–2019. The patient characteristics have been previously reported (Lin et al, 2022). Control samples ($n = 13$) were from victims of traffic accidents with no history or evidence of cardiovascular diseases at autopsy. The study was approved by the Ethics Committee of Helsinki and Uusimaa Hospital District and conducted according to the declaration of Helsinki, and the study subjects gave informed consent.

## Cyclic AMP determination

To measure intracellular cAMP concentrations, H9c2 cells and NMCMs were pretreated with 3-isobutyl-1-methylxanthine (0.1 mM, IBMX, Sigma-Aldrich) for 30 min and then stimulated with the non-selective MC-R agonist α-MSH (0.1 nM) or the MC5-R agonist PG-901 (0.1 nM) for 5, 15, 30, and 60 min. Cells were thereafter lysed with 0.1 M HCl and assayed for cAMP levels with a commercial kit (Cyclic AMP Select ELISA kit, Cayman Chemical, #501040) according to manufacturer's instructions. Results were normalized against total protein concentrations (Pierce™ BCA Protein Assay Kit, ThermoFisher) and expressed as percentage of control samples that were left untreated.

To study concentration responsiveness of the compounds and $G\alpha_i$-evoked inhibition of intracellular cAMP production, a different method using LANCE Ultra® cAMP Detection Kit (PerkinElmer, # TRF0262) was employed. For this purpose, cells were harvested, pipetted into a 96-well OptiPlate (6000 cells/well) and stimulated with different concentrations of α-MSH or PG-901 for 30 min. cAMP levels were determined based on changes in time-resolved fluorescence resonance energy transfer (TR-FRET) signal according to the manufacturer's instructions. In order to detect agonist-induced reduction in cAMP levels, cells were treated with α-MSH or PG-901 in the presence of forskolin (1 μM) or isoprenaline (10 μM).

## Enzyme-linked immunosorbent assays (ELISA) of phosphorylated ERK1/2 and JNK

H9c2 cells and NMCMs were stimulated with the non-selective MC-R agonist α-MSH or the selective MC5-R agonist PG-901 as indicated in the figure legends. Cells were thereafter lysed with Lysis Buffer #6 (R&D Systems) and assayed for the expression levels of phospho-ERK1 (T202/Y204)/ERK2 (T185/Y187) and phospho-JNK with commercial kits (DuoSet IC ELISA, R&D Systems, # DYC1018B & DYC1387B) according to manufacturer's instructions. Results were normalized against total protein concentrations (Pierce™ BCA Protein Assay Kit, ThermoFisher).

## Ca²⁺ mobilization assay

H9c2 cells were seeded into a CellCarrier™ Ultra 96-well plate (8000 cells/well) and were loaded with Fluo-4 Direct™ calcium detection reagent (ThermoFisher, # F10471) in the presence of 5 mM propenicid for 1 h at +37 °C. Baseline was recorded for 40 s before the addition of test compounds followed by a 90 s recording to monitor drug-evoked responses. Ca²⁺ mobilization was measured as increase in fluorescence using Ensight Multimode Plate Reader (PerkinElmer, excitation wavelength 494 nm, emission wavelength 516 nm) and expressed as relative fluorescence units: $\Delta$RFU = (mean of 30 s after drug stimulation-baseline)/baseline. The muscarinic receptor agonist carbachol (100 μM) was used as a positive control.

## ³H-Leucine incorporation assay

To estimate the rate of protein synthesis, cells were cultured in DMEM containing L-[4,5-³H] leucine (1 μCi/ml, PerkinElmer) which incorporates into newly synthetized proteins. Cells were simultaneously treated with either Ang II, ET-1, or Phe and the selective MC5-R agonist PG-901 or with the MC5-R-targeting siRNA. After 24 h, cells were rinsed with PBS and incubated with 10% trichloroacetic acid (TCA) at 4 °C for 30 min to precipitate the proteins. Precipitates were then washed with ice-cold water, lysed with 0.5 M NaOH and mixed with liquid scintillation counter cocktail (Optiphase Supermix, PerkinElmer). Radioactivity in the samples was measured with automatic liquid scintillation counter (Hidex 600 SL, Hidex).

## Histology and immunohistochemistry

Mouse heart samples were cut in the midline of base-apex-axis and fixed in 10% formalin overnight followed by transfer to 70% ethanol and embedding in paraffin and cutting into in 5-μm-thick serial sections. Sections were stained with hematoxylin and eosin (H&E), wheat germ agglutinin (FITC-conjugated, Merck Life Science Oy, UK) and Picro-Sirius red (Abcam, # ab245887) to measure cell size and cardiac fibrosis level in the LV free wall at the level of the papillary muscles. For cell size quantification, at least 100 individual cells with well-defined cell membranes and visible cell nuclei were selected and measured in fields of longitudinally oriented cardiomyocytes. For α-MSH and MC5-R immunohisto-chemistry, sections were incubated in 10 mM sodium citrate buffer (pH 6) for 20 min in a pressure cooker for antigen retrieval. Thereafter, sections were quenched in 1% $H_2O_2$ for 10 min and then blocked in 5% normal horse serum containing 1% BSA. Samples were incubated overnight with a primary antibody against α-MSH (Bioss, MA, USA, # BS-1848R) or MC5-R (Alomone Labs, Jerusalem, Israel, # AMR-025) followed by biotinylated horseradish peroxidase-conjugated secondary antibody incubation and detection with diaminobenzidine (ABC kit, Vector Labs, Burlingame, USA). For isotype control, a consecutive heart section was treated similarly except that the primary MC5-R antibody was replaced by purified normal rabbit IgG (Novus Biologicals, Littleton, CO, USA, # NB810-56910). To quantify apoptotic cells in the heart, one-step TUNEL (Terminal deoxynucleotidyl transferase dUTP nick end labeling) in situ apoptosis assay was performed according to the manufacturer's instructions (eLabscience, Elab Fluor® 647, # E-CK-A324). Sections were counterstained with hematoxylin (CarlRoth) or DAPI, cover-slipped and then scanned with Pannoramic 250 or Pannoramic Midi digital slide scanner (3DHISTECH Kft, Budapest, Hungary). Image analysis was performed using ImageJ software (NIH, Bethesda, MD, USA). Best quality images that most

accurately represent the group mean of measured parameter were selected for Figs. 1H, 6A, 7A and 8A.

## RNA isolation, cDNA synthesis, and quantitative RT-PCR

H9c2, NMCMs, and hiPSC-CMs were collected into QIAzol Lysis Reagent or Trizol Reagent (Invitrogen) and total RNA was extracted using Direct-zol RNA Miniprep and Microprep (Zymo Research, CA, USA), respectively. Heart samples were first homogenized in QIAzol Lysis Reagent using the Qiagen TissueLyser LT Bead Mill (QIAGEN, Venlo, Netherlands) and total RNA was thereafter extracted using Direct-zol RNA Miniprep. RNA was reverse-transcribed to cDNA (PrimeScript RT reagent kit, Takara Clontech) and quantitative real-time polymerase chain reaction (RT-PCR) was performed with SYBR Green protocols (Kapa Biosystems, MA, USA) and a real-time PCR detection system (Applied Biosystems 7300 Real-Time PCR system) (Rinne et al, 2017; Kadiri et al, 2021). Target gene expression was normalized to a housekeeping gene (ribosomal protein S18; RPS18, glyceraldehyde-3-phosphate dehydrogenase; GAPDH or β-actin; ACTB) using the comparative ΔCt method and results are presented as relative transcript levels ($2^{-\Delta\Delta Ct}$). Primer sequences are presented in Appendix Tables S4–S6.

## Western blot

Cell and heart samples were lysed in RIPA buffer supplemented with protease and phosphatase inhibitors (Complete Mini, Roche and Halt™ Phosphatase Inhibitor Cocktail, ThermoFisher). Aliquots of total protein were separated by SDS-PAGE and transferred to a nitrocellulose membrane. After blocking with 5% nonfat milk, membranes were incubated with primary antibodies overnight at 4 °C. The following primary antibodies were used: anti-MC5-R (Alomone Labs, Jerusalem, Israel, # AMR-025), anti-α-MSH (Bioss, MA, USA, # BS-1848R), anti-phospho-JNK (Cell Signaling Tech, # 4668), anti-JNK (R&D Systems, # AF-1387), anti-phospho-ERK1/2 (Cell Signaling Tech, # 9107), anti-ERK1/2 (Cell Signaling Tech, # 9106), anti-phospho-p38 (Cell Signaling Tech, # 9215), anti-p38 (Cell Signaling Tech, # 9212), anti-phospho-AMPKα (Cell Signaling Tech, # 2535), anti-AMPKα (Cell Signaling Tech, # 2532), anti-phospho-Akt (R&D Systems, # AF887), anti-Akt (R&D Systems, # MAB2055), anti-phospho-CREB (ThermoFisher, # MA5-11192), anti-CREB (Cell Signaling Tech, #9197) and anti-NT-proBNP (Abcam, # ab13115) antibody. Thereafter, membranes were washed and incubated with horseradish peroxidase-conjugated anti-IgG secondary antibody (Cell Signaling Tech, Frankfurt, DE) followed by detection using a chemiluminescence system (Pierce™ ECL Western Blotting Substrate, ThermoFisher) and Sapphire Biomolecular Imager (Azure Biosystems). The results for target protein expression were normalized to β-actin (Merck Life Science Oy, # 2066) or vinculin (BioRad, # MCA465GA) expression to correct for loading.

## Statistics

Statistical analyses were performed with GraphPad Prism 8 software (La Jolla, CA, USA). Statistical significance between the experimental groups was determined by unpaired Student's t-test, one-way ANOVA followed by Dunnett post hoc tests or two-way ANOVA followed by Bonferroni post hoc tests. Quantitative PCR data from hiPSC-CMs was analyzed using ΔCt-values and randomized block ANOVA (individual experiments and treatment as factors) as previously described (Karhu et al, 2021). Pearson correlation coefficients were calculated for gene associations. The D'Agostino and Pearson omnibus normality test method was employed to test the normality of the data. Possible outliers in the data sets were identified using the regression and outlier removal (ROUT) method at Q-level of 1%. Data are expressed as mean ± standard error of the mean (SEM). Results were considered significant for $P < 0.05$.

## Data availability

This study includes no data deposited in external repositories.

## Peer review information

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

## Acknowledgements
We thank Sanna Bastman, Hanna Haukkala, Satu Mäkelä, Johanna Jukkala, and Annika Korvenpää for their excellent technical support. The histological methods were performed by the Histology core facility of the Institute of Biomedicine, University of Turku, Finland. This work was financially supported by grants from the Research Council of Finland (grant 315351 to PR, grant 266621 to HR, grant 321564 to VT, and grant 333284 to RK), the Sigrid Jusélius Foundation (to PR, HR, VT, and RK), the Finnish Cultural Foundation (to PR and LP), Drug Research Doctoral Programme (to AS), the Finnish Foundation for Cardiovascular Research (to LP, HR, VT, RK, and PR), the Instrumentarium Science Foundation (to AS), and the National Institutes of Health (grant GM-104080 to MC).

## Author contributions
**Anni Suominen**: Conceptualization; Formal analysis; Investigation; Methodology; Writing—original draft; Writing—review and editing. **Guillem Saldo Rubio**: Formal analysis; Investigation; Writing—review and editing. **Saku Ruohonen**: Formal analysis; Investigation; Writing—review and editing. **Zoltán Szabó**: Formal analysis; Investigation; Writing—review and editing. **Lotta Pohjolainen**: Investigation; Writing—review and editing. **Bishwa Ghimire**: Formal analysis; Writing—review and editing. **Suvi T Ruohonen**: Investigation; Writing—review and editing. **Karla Saukkonen**: Investigation; Writing—review and editing. **Jani Ijas**: Investigation; Writing—review and editing. **Sini Skarp**: Formal analysis; Investigation; Writing—review and editing. **Leena Kaikkonen**: Investigation; Writing—review and editing. **Minying Cai**: Funding acquisition; Methodology; Writing—review and editing. **Sharon L Wardlaw**: Formal analysis; Methodology; Writing—review and editing. **Heikki Ruskoaho**: Supervision; Funding acquisition; Methodology; Writing—review and editing. **Virpi Talman**: Supervision; Funding acquisition; Methodology; Writing—review and editing. **Eriika Savontaus**: Funding acquisition; Writing—review and editing. **Risto Kerkelä**: Supervision; Funding acquisition; Methodology; Writing—review and editing. **Petteri Rinne**: Conceptualization; Formal analysis; Supervision; Funding acquisition; Investigation; Methodology; Writing—original draft; Writing—review and editing.

## Disclosure and competing interests statement
The authors declare no competing interests.

# Expanded View Figures

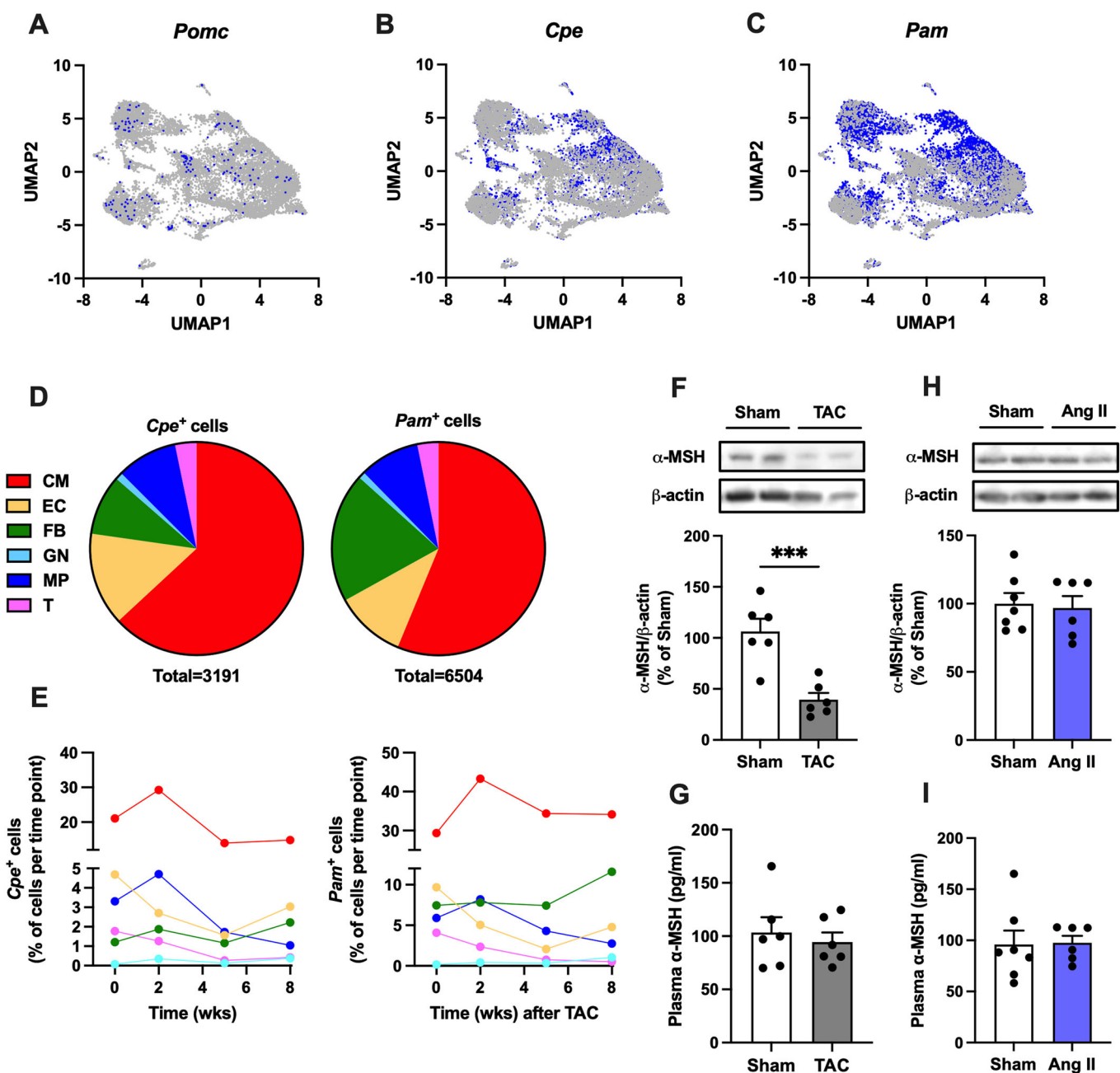

**Figure EV1. Single-cell analysis of *Pomc*-, *Cpe*-, and *Pam*-expressing cells in the heart of pressure-overloaded mice.**

(A–C) Uniform Manifold Approximation and Projection (UMAP) showing 11,492 single cells isolated from C57Bl mice at different stages of cardiac hypertrophy. Blue dots indicate the localization of pro-opiomelanocortin (*Pomc*)-, carboxypeptidase E (*Cpe*)-α-amidating monooxygenase (*Pam*)-expressing cells in the UMAP-plot. CM indicates cardiomyocyte; EC, endothelial cell; FB, fibroblast; GN, granulocyte; MP, macrophage; and T, T cell. (D) Pie charts showing the relative distribution of *Cpe*+- and *Pam*+ cells in each cell type. (E) Changes in the relative amount of *Cpe*+ and *Pam*+ cells in each cell type as a function of time after transverse aortic constriction (TAC) surgery. *Pam*+ and *Cpe*+ cells are expressed as percentage of total number of sequenced at each time point. (F) Representative Western blots and quantification of α-MSH (normalized to β-actin) in the LV samples of sham- and TAC-operated mice. $n = 6$ mice per group. ***$P < 0.001$ by Student's t test. (G) α-MSH concentration in the plasma of sham- and TAC-operated mice 5 weeks after the surgery. $n = 6$ mice per group. (H) Representative Western blots and quantification of α-MSH (normalized to β-actin) in the LV samples of sham-operated and Ang II-infused (4 weeks) mice. $n = 7$ in sham, $n = 6$ in Ang II. (I) α-MSH concentration in the plasma of sham-operated and Ang II-infused (4 weeks) mice. $n = 7$ in sham, $n = 6$ in Ang II. Data information: Data are mean ± SEM, each dot represents individual mouse.

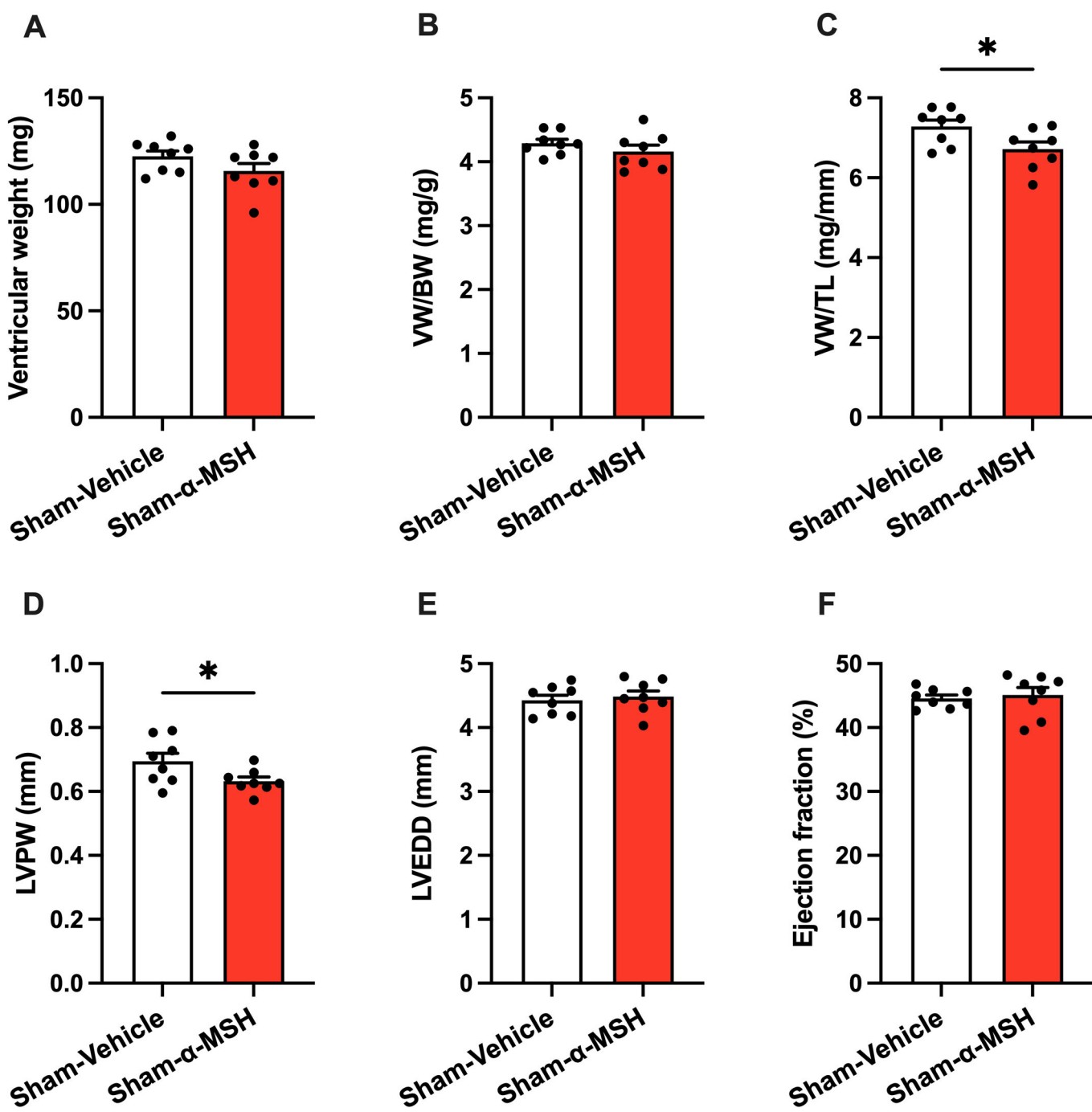

**Figure EV2. Ventricular weight and echocardiography in vehicle- and α-MSH-treated mice after 8 weeks of sham operation.**

(A–C) Ventricular weight, ventricular weight to body weight ratio (VW/BW) and ventricular weight to tibia length ratio (VW/TL) in sham-operated mice treated with either vehicle or α-MSH analog (melanotan II; MT-II). (D–F) Left ventricular posterior wall thickness (LVPW), left ventricular end-diastolic dimension (LVEDD), and ejection fraction analyzed by echocardiography at the end of the experiment. Data information: Data are mean ± SEM, $n = 8$ in sham-vehicle and $n = 8$ in sham-α-MSH. *$P < 0.05$ by Student's t test.

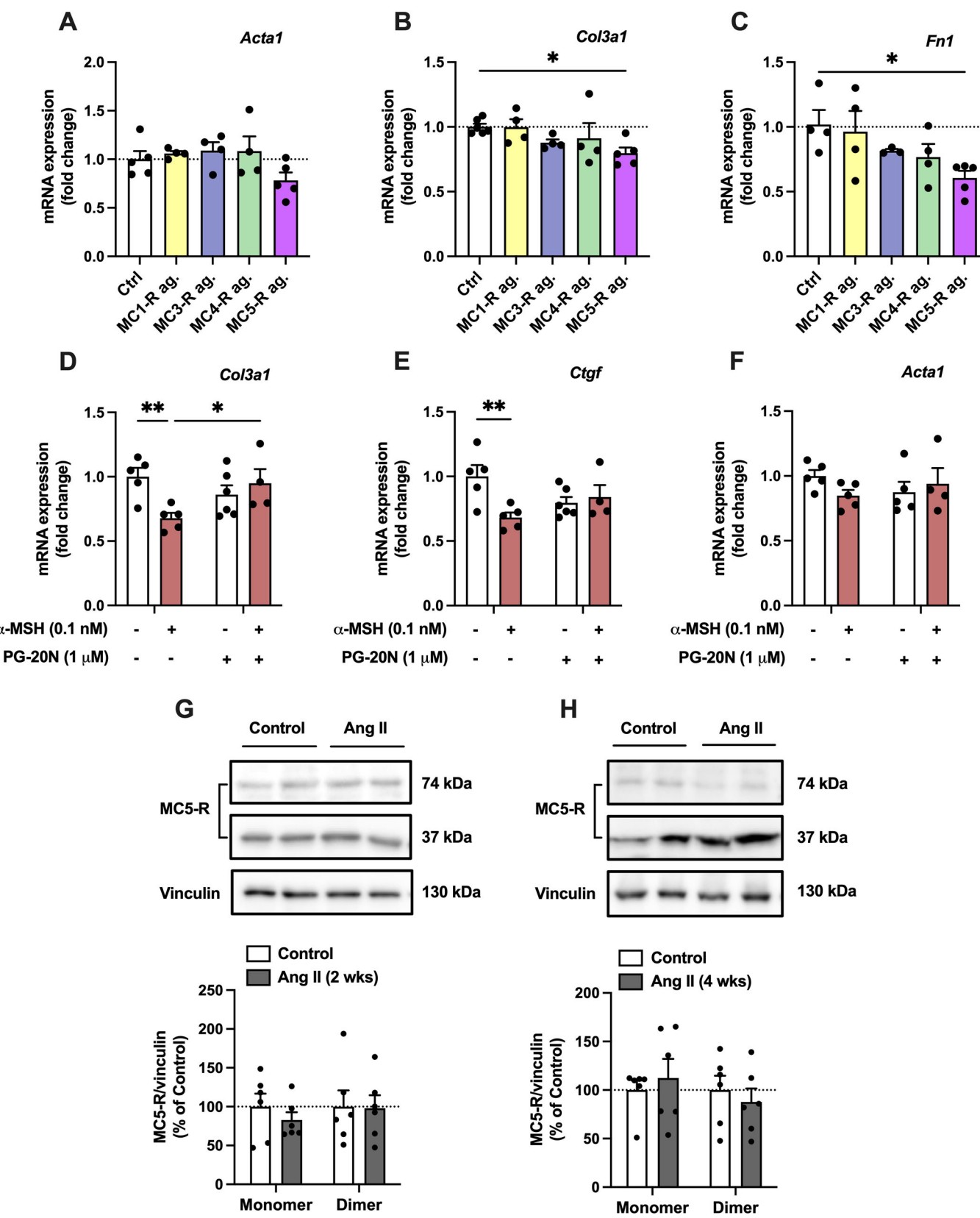

**Figure EV3. The effects of subtype selective MC-R agonists on gene expression in H9c2 cells and MC5-R expression in hypertrophied mouse heart.**

(A–C) *Acta1*, *Col3a1*, and *Fn1* mRNA expression in H9c2 cells treated with MC1-R, MC3-R, MC4-R, or MC5-R selective agonist (10 nM for all agonists) for 3 h. $n = 4$–6 per group (technical replicates) in each graph from 2 independent experiments. (D–F) *Nppb*, *Acta2*, and *Tgfb1* expression in H9c2 cells treated with α-MSH (0.1 nM) for 1 h in the absence or presence of the selective MC5-R antagonist PG-20N (1 μM). $n = 4$–6 per group (technical replicates) in each graph from 2 independent experiments. (G, H) Representative Western blots and quantification of MC5-R monomer and dimer forms (normalized to vinculin) in the LV samples of control, sham-operated mice and Ang II-infused mice. Ang II was infused for 2 (G) or 4 weeks (H). $n = 5$–6 mice (biological replicates) per group in each graph. Data information: Data are mean ± SEM, $n = 4$–6 per group, *$P < 0.05$ and **$P < 0.01$ for the indicated post hoc comparisons by 1-way ANOVA and Dunnett post hoc tests (B, C) or by 2-way ANOVA and Bonferroni post hoc tests (D, E).

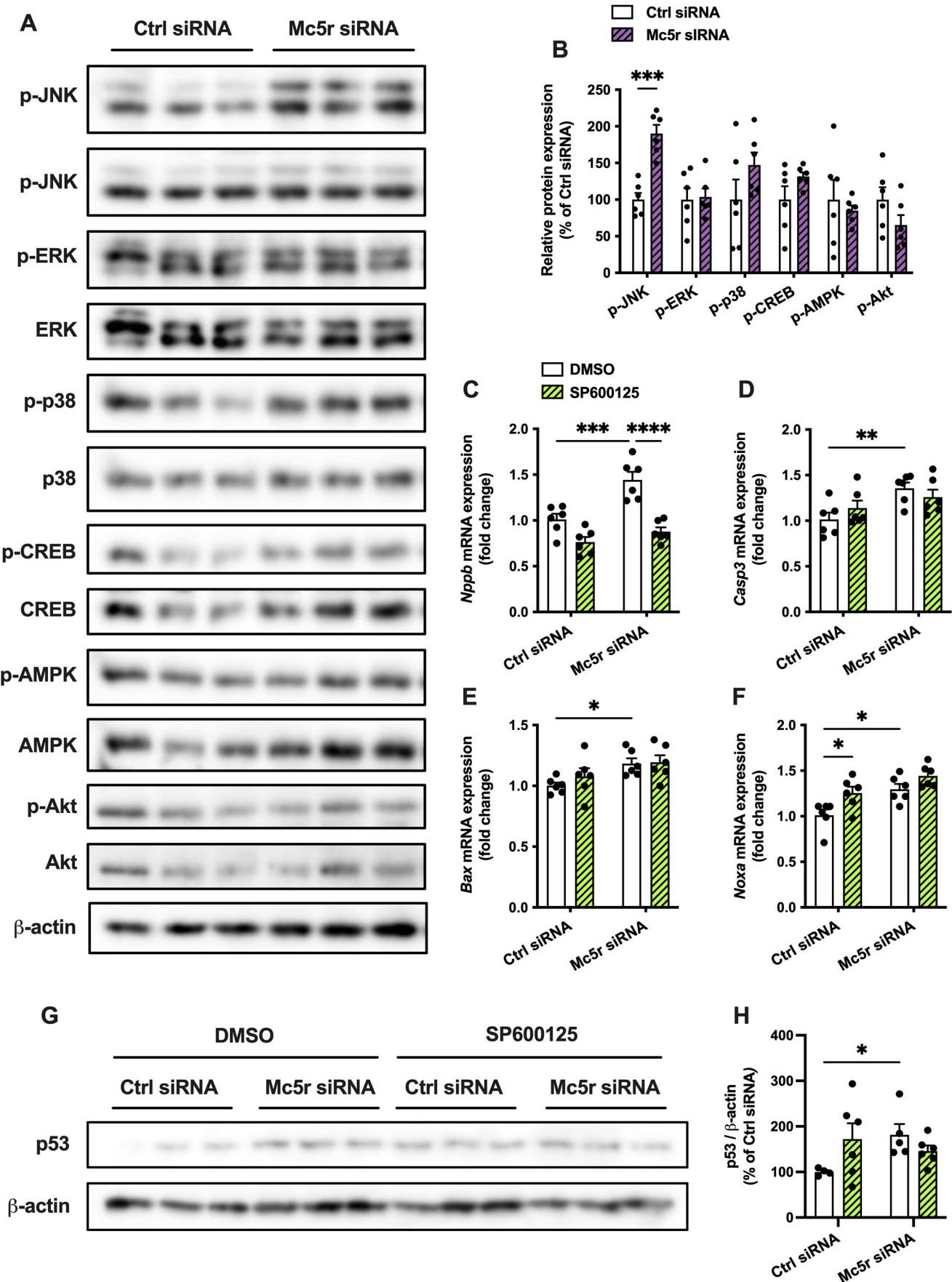

◀ **Figure EV4. Mc5r knockdown enhances the expression of pro-apoptotic markers in neonatal mouse ventricular cardiac myocytes (NMCM).**

(A, B) Representative Western blots and quantification of p-JNK (normalized to total JNK), p-ERK (normalized to total ERK), p-p38 (normalized to total p38), p-CREB (normalized to total CREB), p-Akt (normalized to total Akt) and p-AMPK (normalized to total AMPK) in H9c2 cells treated transfected with control siRNA or *Mc5r*-targeting siRNA for 24 h. $n = 6$ (technical replicates) per group from 2 independent experiments. (C–F) qPCR analysis of *Nppb, Casp3, Bax,* and *Noxa* mRNA expression in NMCMs treated with or without the JNK inhibitor SP600125 (10 μM) for 30 min followed by transfection with control siRNA or *Mc5r* targeting siRNA for 24 h. $n = 6$ (technical replicates) per group from 2 independent experiments. (G, H) Representative Western blots and quantification of p53 (normalized to β-actin) in NMCMs treated with or without the JNK inhibitor SP600125 (10 μM) for 30 min followed by transfection with control siRNA or *Mc5r* targeting siRNA for 24 h. $n = 6$ (technical replicates) per group from 2 independent experiments. Data information: Data are mean ± SEM. *$P < 0.05$, **$P < 0.01$, ***$P < 0.001$, and ****$P < 0.0001$ for the indicated post hoc comparisons by 2-way ANOVA and Bonferroni post hoc tests (C–H).

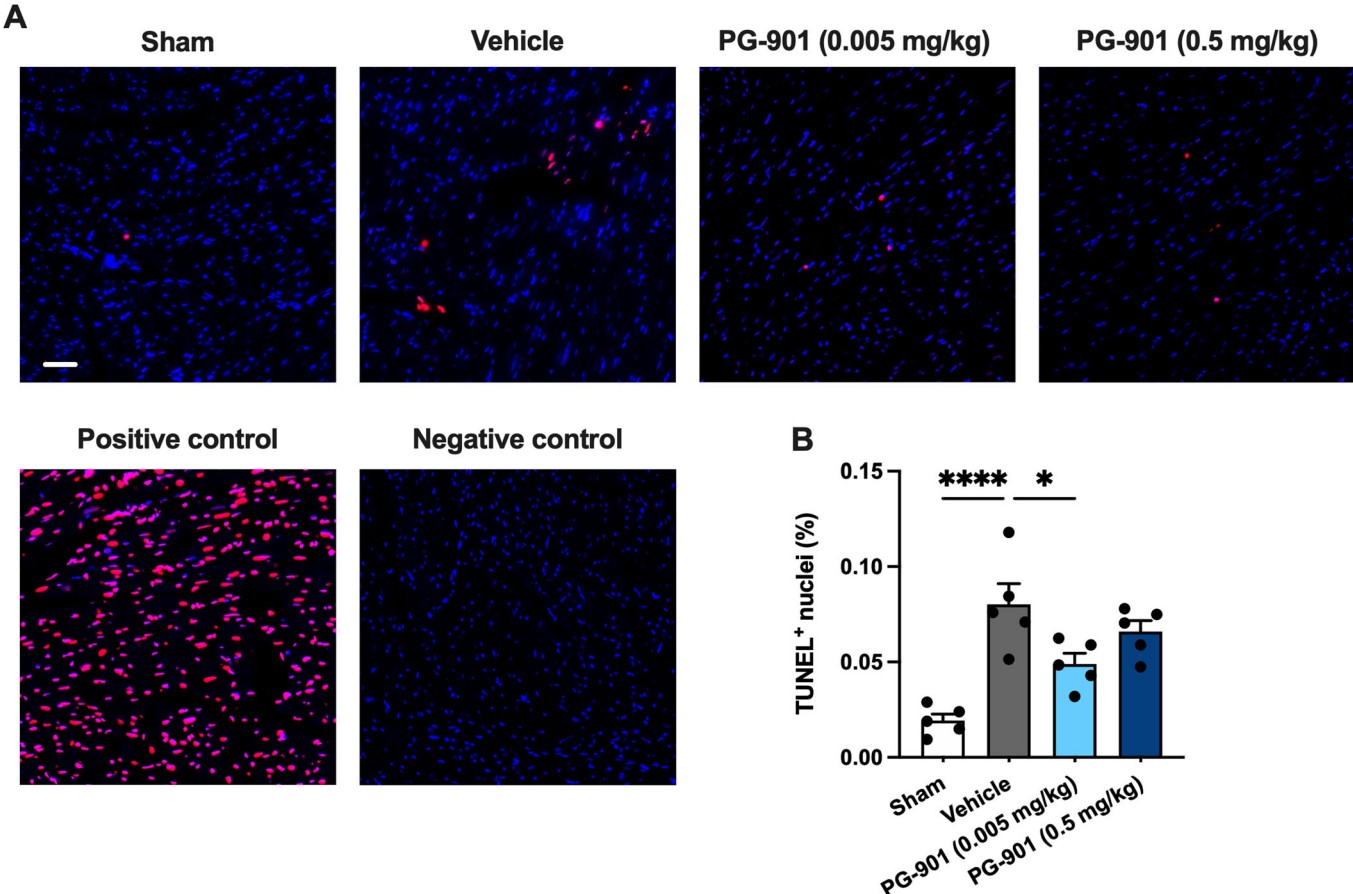

**Figure EV5. MC5-R activation with PG-901 reduced the number of apoptotic cells in the heart of TAC-operated mice.**

(A, B) Representative images and quantitative analysis showing relative amount of apoptotic TUNEL-positive nuclei in the LV of sham- and TAC-operated in sham- and TAC-operated mice treated with either vehicle or PG-901 (0.5 or 0.005 mg/kg/day). Scale bar, 50 μm. Positive control was treated with DNase I (0.2 U/μl, 15 min) before being subjected to TUNEL assay. Negative control was treated in a similar way as experimental samples but the labeling was carried out in the absence of TdT enzyme. Data information: Data are mean ± SEM, $n = 5$ mice per group. $*P < 0.05$ and $****P < 0.0001$ for the indicated comparisons by 1-way ANOVA and Dunnett's post hoc tests.

