## [Peer Review File · EMBO Reports]

α -Melanocyte-stimulating hormone alleviates pathological cardiac remodeling via melanocortin 5 receptor

Anni Suominen, Guillem Saldo Rubio, Saku Ruohonen, Zoltan Szabo, Lotta Pohjolainen, Bishwa Ghimire, Suvi Ruohonen, Karla Saukkonen, Jani Ijas, Sini Skarp, Leena Kaikkonen, Mingying Cai, Sharon Wardlaw, Heikki Ruskoaho, Virpi Talman, Eriika Savontaus, Risto Kerkelä, and Petteri Rinne

Corresponding author(s): Petteri Rinne (pperin@utu.fi)

Review Timeline:

Transfer Date:	19th Jan 24
Editorial Decision:	22nd Jan 24
Revision Received:	23rd Jan 24
Accepted:	16th Feb 24

Editor: Deniz Senyilmaz Tiebe

Transaction Report: A revised version of this manuscript was transferred to EMBO reports following peer review at EMBO Molecular Medicine.

EMM-2023-18025

Title: α -Melanocyte-stimulating hormone regulates pathological cardiac remodeling via melanocortin 5 receptor

Corresponding author: Dr. Petteri Rinne

Authors' response to reviewers

We thank the editors and reviewers for their positive feedback and constructive comments, which greatly helped us to strengthen the manuscript and its conclusions. We have now addressed the reviewers' concerns, and carefully and thoroughly revised our paper. All changes that were made to the manuscript are highlighted in the 'Manuscript with tracked changes'-file.

Referee #1:

Suominen et al. investigated the effects of α -Melanocyte-stimulating hormone on cardiac remodeling after experimental pressure overload. The authors show that α -MSH is expressed mainly in cardiomyocytes in mouse hearts, and becomes downregulated after chronic pressure overload by TAC. The authors demonstrate that supplementation of α -MSH inhibits cardiac hypertrophy after TAC in mice, mainly through acting on the MC5R receptor, which is upregulated in mice after TAC and in human failing hearts. Cardiomyocyte specific knock-out mice of this MC5R receptor promoted mildly exaggerated hypertrophy and systolic cardiac dysfunction, and more fibrosis. Activation of the MC5R receptor by the agonist PG-901 in mice in vivo after TAC did not lead to reduced cardiac hypertrophy on the organ level, but mildly reduced fibrosis and reduced systolic and diastolic heart dysfunction. Mechanistically, the authors suggest that α -MSH acts on cardiomyocytes via reducing JNK activation and intracellular cAMP generation, however this remains vague and inconclusive.

In summary, the concept laid out in the paper is novel and the in vivo data in mice are largely convincing, although some inconsistencies exist.

Response: We thank the reviewer for the insightful and constructive comments, and we are pleased to read that the manuscript has raised considerable interest.

However, the molecular mechanism how α -MSH might act on the heart remains completely unclear. The authors have to address the following points:

1) How does α -MSH or the MC5R receptor agonist reduce cardiac hypertrophy and fibrosis? This remains unclear. I am not convinced that it is via alterations in JNK activation. Some unbiased approach might be needed here (phosphoproteomics etc.). If the authors think that it is JNK, a JNK inhibitor should be applied to MC5-R knock-out mice after TAC, to see whether the phenotype can be rescued.

Response: We thank the reviewer for these comments and acknowledge that further mechanistic experiments to investigate the dependency of the observed mouse phenotypes on the JNK pathway would be valuable. Unfortunately, we have discontinued the maintenance of Mc5r-cKO mouse line due to breeding problems (-> only cryopreserved sperm available) and performing new experiments with this mouse model would not be possible within a reasonable time frame.

Instead, we performed a wider screen for phosphoproteins in PG-901-treated H9c2 cells. We selected targets that are known to be modulated by melanocortin 5 receptor signaling in other cells/models (*Xu Y et al., Cell Mol Life Sci. 2020 Oct;77(19):3831-3840*), and quantified these by Western blotting. As shown below (**Figure R1**), PG-901 did not affect the phosphorylation status of CREB (**B**), ERK (**C**), p38 (**D**), Akt (**E**) or AMPK (**F**).

Figure R1. The effects of the MC5-selective agonist PG-901 on intracellular signaling pathways in H9c2 cells. (A through F) Representative Western blots and quantification of p-CREB (normalized to total CREB), p-ERK (normalized to total ERK), p-p38 (normalized to total p38), p-Akt (normalized to total Akt) and p-AMPK (normalized to total AMPK) in H9c2 cells treated with PG-901 (0.1 nM) for 5-60 minutes. Data are mean \pm SEM, n=4-6 per group.

We also did the same screening for samples from *Mc5r*-silenced H9c2 cells (**Extended Figure 4A** in the revision). The only significant change was observed in JNK phosphorylation, which was increased in *Mc5r*-silenced cells (**Extended Figure 4B**). We also performed a new experiment with primary cardiomyocytes (NMCs) to investigate whether the induction of *Nppb* expression is similarly dependent on JNK signaling as observed in H9c2 cells (**Figure 4M**). Indeed, JNK inhibition with SP600125 completely reversed the upregulation of *Nppb* in *Mc5r*-silenced NMCs (**Extended Figure 4C**), further consolidating the link between MC5-R signaling, JNK modulation and cardiomyocyte hypertrophy.

However, based on the existing and conflicting literature on the role of JNK in cardiac hypertrophy (reviewed in *Physiol Rev.* 2010 Oct;90(4):10.1152/physrev.00054.2009), it would be highly challenging to prove the contribution of the JNK pathway to the observed phenotype *in vivo*. Earlier findings indicate that JNK activation is a dynamic and transient signaling event (peaking at ~30 min) after applying pressure overload and that different JNK isoforms might have separate and non-redundant roles in the process. Although *in vitro* studies strongly argue for a prohypertrophic role of JNKs, loss-of-function approaches to silence JNK or its upstream regulators *in vivo* have resulted in promotion as well as attenuation of cardiac growth response (as discussed on manuscript **page 15, lines 531-539**). Furthermore, JNK activation *in vivo* (in transgenic animal models) does not induce cardiac hypertrophy.

Therefore, we have been careful not to claim that the observed *in vivo* phenotypes would be dependent on the JNK pathway, and disclose the uncertainty related to the involvement of JNK signaling in the observed *in vivo* phenotypes (manuscript **page 15, lines 535-539**): “*In vitro* experiments in the current study demonstrate that MC5-R regulates hypertrophic growth of cardiomyocytes in a JNK-dependent manner. However, further studies are warranted to determine whether MC5-R-induced JNK reduction has a direct effect on cardiomyocyte growth *in vivo*”.

Regarding the mechanistic insight of the manuscript, we think that the most important finding is the demonstration that α -MSH evokes an anti-hypertrophic effect by activating MC5-R in cardiomyocytes. From a drug development perspective, this is the key mechanistic finding revealing that cardiomyocytes are the main effector cells (among other cell types of the heart) and that MC5-R is the target receptor (among MC-R subtypes) for the anti-hypertrophic and -fibrotic effect of α -MSH.

2) Figure 1A: instead of hypothalamus, the levels in the pituitary should be shown, where α -MSH is mainly produced.

Response: As requested by the reviewer, we measured α -MSH concentration in the pituitary gland and present this new data in **Figure 1A** (instead of hypothalamic α -MSH level).

3) Figure 1H: Fibrosis should be measured in hearts by histology.

Response: We fully agree with the reviewer and quantified the level of LV fibrosis in α -MSH-treated mice. Consistent with the phenotype observed in Mc5r-cKO mice and PG-901-treated mice, TAC-induced fibrosis was significantly attenuated by chronic α -MSH treatment (**Figure 1H & 1N**).

4) How is fibrosis regulated by α -MSH? Could there be a cross talk of myocytes to fibroblasts? More mechanistic insight is clearly needed.

Response: We thank the reviewer for this comment. We found that α -MSH downregulated fibrotic markers such as TGF β and CTGF in cultured cardiomyocytes, which are known to promote cardiac fibrosis through humoral and growth factor-mediated pathways.

We are aware that in the context of cardiac hypertrophy, cardiac myocyte-to-cardiac fibroblast (CM-to-CF) and CF-to-CM crosstalk significantly contribute to pathological cardiac remodeling. Mice with CM-specific gene manipulation might have a significant effect on cardiac fibrosis without any clear signs of cardiac hypertrophy. For instance, mice with CM-specific overexpression of CTGF show enhanced cardiac fibrosis after TAC-induced pressure overload but are not sensitized to the development of cardiac hypertrophy (Yoon PO et al., *J Mol Cell Cardiol.* 2010 Aug;49(2):294-303). This might have also relevance to our findings of downregulated CTGF expression in PG-901 treated cells and in TAC-operated mice as well as upregulated CTGF expression in Mc5r-silenced cardiomyocytes.

Against this background, it is very likely that α -MSH regulates fibrosis, at least in part, through CM-to-CF cross talk. However, since we identified cardiac myocytes as the main effector cells for α -MSH-mediated anti-hypertrophic and -fibrotic regulation, we feel that it is beyond the scope of this manuscript to investigate cross talk between CM and CF in this regard. Particularly, taking into account that the mechanisms of CM-to-CF cross talk are not fully understood and involve e.g. release of paracrine factors, direct cell-cell interactions and cell interaction with the extracellular matrix (Zhang P & U Mende JS, *Am J Physiol Heart Circ Physiol.* 2012 Dec 15;303(12):H1385-96), it would be challenging and time-consuming to start addressing these possibilities.

5) P-JNK Western blots need to be analyzed in the myocardium in the mouse models shown (Figures 1, 6, 7, 8).

Response: As requested by the reviewer, we analyzed cardiac p-JNK levels in our mouse models. First, we analyzed the LV samples from cardiomyocyte-specific Mc5r-cKO mice for p-JNK levels as well as for the phosphorylation level of other MAP kinases (ERK and p38) and CREB. The results of the WB analysis are shown below (**Figure R2**). Sham- or TAC-operated Mc5r-cKO mice did not show any alteration in p-JNK (**B**), p-ERK (**C**), p-p38 (**D**) or p-CREB (**E**) level compared to control mice.

Figure R2. Representative Western blots and quantification of p-JNK, p-ERK, p-p38 and p-CREB (normalized to β -actin) in sham- and TAC-operated *Mc5r^{fl/fl}*, MYH6-MCM and *Mc5r-cKO* mice. Data are mean \pm SEM, $n=3-7$ per group.

Second, we analyzed the LV samples from PG-901-treated C57Bl/6N mice for p-JNK, p-ERK and p-p38 levels (**Figure R3**).

Figure R3. Representative Western blots and quantification of p-JNK, p-ERK and p-p38 in sham- and TAC-operated C57Bl/6N mice treated with either vehicle or PG-901 (0.5 or 0.005 mg/kg/day). Data are mean \pm SEM, n=6-8 per group. ** $P < 0.01$ and *** $P < 0.001$ for the indicated comparisons by 1-way ANOVA and Dunnett post hoc tests.

PG-901-treatment did not significantly change the expression of these phosphoproteins in TAC-operated C57Bl/6N mice. There was however a tendency for lower p-JNK level ($P=0.09$ by 1-way ANOVA and Dunnett's post hoc test) in TAC mice treated with 0.5 mg/kg of PG-901. Overall, it is difficult to draw any valid conclusion from these results. First, the samples from these experiments were already relatively old, which caused challenges in detecting clear bands for the phosphorylated forms of these proteins. Second, the signal for e.g. p-JNK in these Western blots is derived from both cardiomyocytes and non-cardiomyocytes such as fibroblasts, endothelial cells and leukocytes, thus possibly diluting the effect that might be occurring in cardiomyocytes.

These results imply that PG-901 or MC5-R deficiency does not affect the expression p-JNK at the investigated time points after TAC operation, but taking into account the uncertainties related to these analyses, we would rather omit these data from the manuscript. Since the expression levels of the other phosphoproteins were also unaffected, the results do not help in delineating the signaling mechanisms behind the observed *in vivo* phenotypes.

6) Figure 4: H9c2 cells are not a good model for cardiomyocyte hypertrophy at all.

Response: We agree with the reviewer and have therefore used neonatal mouse ventricular cardiac myocytes (NMCM) in parallel with H9c2 cells. To comply with the 3R principles, we first aimed to identify α -MSH-mediated responses in H9c2 cells and then validated the key findings in NMCMs. Since the main signaling responses appeared similarly in H9c2 cells and NMCMs, we relied more on the H9c2 model when performing subsequent mechanistic experiments to reduce the number of mice. Based on the reviewer's comment, we have performed new experiments with NMCMs to further demonstrate that the MC5-R mediated effects are appearing similarly in both *in vitro* models (**Extended Figure 3 & Figure R5**).

7) Figure 5: α -MSH should be measured in failing human hearts. The effects of PG-901 on cardiac gene-expression are extremely small. Cell size should be measured.

Response: Measurement of α -MSH level in failing human hearts would be indeed interesting and relevant. Based on the increased *POMC* expression in the human cardiomyopathy patient samples, it is expected that α -MSH expression is similarly increased. This hypothesis is also supported by the clinical finding of increased α -MSH concentration in the plasma of cardiomyopathy patients (*Yamaoka-Tojo M et al, Intern Med. 2006;45:429–434*). However, due to the lack of respective protein samples from the LV, this hypothesis cannot be tested.

We agree that the effects of PG-901 on gene expression in hiPSC-CMs are extremely small and therefore measurement of cell size could provide further evidence on the role of MC5-R in modulating hypertrophic responses in these cells. However, based on our previous experiments, cell size, as measured by cross-sectional area, does not significantly change in response to ET-1 treatment, although *Nppa* and *Nppb* are robustly upregulated. There might be an increase in cell volume in ET1-treated hiPSC-CMs (Johansson M *et al.*, *Biol Open*. 2020 Sep 21;9(9):bio052381), but we do not have a validated protocol for this purpose. Therefore, instead of cell size or volume, we quantified NT-proBNP protein expression, which could be used a molecular marker of cell hypertrophy (Pohjolainen L *et al.*, *Front Pharmacol*. 2021 Jan 20;11:553852) and is considered as the gold standard biomarker in determining the diagnosis and prognosis of human heart failure. We observed that PG-901 treatment attenuated the hypertrophic effect of ET-1 in hiPSC-CMs (**Figure 5H, manuscript page 11, lines 355-357**).

8) Figure 8: Why is there no anti-hypertrophic effect of PG-901 in vivo?

Response: The lack of effect of PG-901 on hypertrophic response in TAC-operated C57Bl/6N might relate to the controversial role of JNK signaling in pathological cardiac hypertrophy *in vivo* (as discussed above; response to comment 1 and on manuscript **page 15, lines 531-535**). Therefore, the impact of JNK inhibition on the hypertrophic response could be sensitive to nuances in the experimental settings (*e.g.* C57Bl/6J vs C57Bl/6N, time point and timing of dosing).

However, we consider that the most likely explanation is that PG-901 evokes off-target effect(s) in cardiomyocytes and/or non-myocytes that are counteracting/masking the possible anti-hypertrophic effect that is mediated *via* MC5-R. This issue is more thoroughly discussed in response to comment 11 by Referee #3 (page 17 of this response letter) and on **manuscript page 14/lines 507-514**.

Referee #2

In this study, Suominen et al. focused on alpha-melanocyte-stimulating hormone (α-MSH) in cardiac remodeling under pressure overload. The authors found that α-MSH was actually highly expressed in the heart, compared to its conventional expression location, the hypothalamus. They moved to showed that cardiomyocytes showed high expression of α-MSH. Importantly, α-MSH treatment ameliorated cardiac remodeling under pressure overload. At the molecular level, the authors found that α-MSH suppressed JNK signaling, which was accompanied by downregulation of a number of hypertrophy markers. In addition, they showed that MC5-R is the receptor for α-MSH, suggesting α-MSH exerts a paracrine function in the heart. Finally, MC5-R deletion mice displayed exacerbated pathological remodeling in response to pressure overload. Overall, the experiments were thoughtfully designed and nicely executed. I have a few comments for the authors to consider.

Response: We thank the reviewer for the constructive and insightful comments.

1. The authors nicely tested several different stimuli of hypertrophy and α -MSH signaling. However, the data are somewhat conflicting. In figure 2, α -MSH reduced angiotensin II-induced gene expression for *Nppb* and *Acta 2*, *pJNK* level, and leucine incorporation. However, angiotensin II infusion did not affect the dimerization of MC5-R. Does this mean that angiotensin II does not affect the activity/expression of MC5-R but reducing α -MSH? Is this the mechanism of angiotensin II toxicity? Have the authors tested the level of α -MSH in circulation and/or in the heart under angiotensin II infusion?

Response: We thank the reviewer for these comments and acknowledge that the results obtained from Ang II -models might appear confusing/conflicting. We have discussed these discordant findings on **page 16/lines 563-569** of the revised manuscript and speculate that the difference might be caused by the dominant influence of noncardiomyocyte cells such as fibroblasts as mediators of Ang II-induced hypertrophic response *in vivo*. Vasoconstriction and consequent increase in blood pressure also contribute to cardiac hypertrophy in Ang II-infused mice. Accordingly, the mechanisms of Ang II-induced cardiomyocyte hypertrophy differ between *in vitro* and *in vivo* settings, thus possibly explaining why MC5-R activation/inactivation evokes a clear phenotype only in cultured cardiomyocytes.

As suggested by the reviewer, we also measured the level of α -MSH in the heart/LV and plasma of Ang II-infused mice. Consistent with the earlier findings of unchanged cardiac MC5-R expression, Ang II-infusion for 4 weeks did not affect circulating or cardiac α -MSH levels (**Extended Figure 1H and I**). Therefore, it appears that the α -MSH/MC5-R axis responds in a cohesive way to different hypertrophic stimuli, *i.e.* reduced α -MSH and MC5-R expression in TAC-operated mice, while no effect is observed in Ang II-infused mice.

2. Across the entire study, most if not all differences are small, yet significant. While I do not doubt the significance of the difference and the implication of the work, it is striking to me regarding the small differences. Does this mean that α -MSH signaling is not major? I would recommend some explanation/discussion from the authors. Even better, the authors could compare this phenotype with findings from other similar hormones.

Response: The reviewer raises an important point, which should be discussed in the manuscript. It might well be that MC5-R signaling does not play a major role in cardiac remodeling. However, although we have not done head-to-head comparison in our experiments, it appears that the effect of α -MSH is stronger compared to the MC5-R agonist in both *in vitro* and *in vivo* models. This raises a possibility that PG-901 is a partial or biased agonist of MC5-R and therefore, does not fully recapitulate the actions of α -MSH. PG-901 might also evoke off-target effects that mask the primary effect occurring through MC5-R activation (more detailed discussion below/response to comment 3).

It is also well-established that melanocortin receptors, like other G protein-coupled receptors, undergo agonist-mediated desensitization and internalization, which becomes particularly dominant after repeated, long-term treatment with an MC-R agonist. In any case, we want to emphasize that PG-901 was only used as pharmacological tool to further dissect the role of MC5-R in cardiac remodeling. Further experiments are clearly warranted to design new MC5-R agonists and test their effectiveness in similar disease models. This would better allow addressing the question whether MC5-R signaling is major in cardiac remodeling or not, and whether targeting MC5-R holds promise as a therapeutic strategy in managing heart failure.

3. Figure 8, PG-091 did not affect cardiac hypertrophy but improve cardiac function under pressure overload. This finding is not consistent with the *in vitro* results in figure 4E&H. Any explanations?

Response: We have to acknowledge that there is clearly something with PG-901 that we have not figured out. For example, in hiPSC-CMs, PG-901 evokes a significant and dose-dependent effect on *NPPB* expression, but the direction of effect varies from experiment to experiment (**Figure R4**). Basal *MC5R* expression, which varies between different batches of hiPSC-CM differentiation, might explain this confusing finding. PG-901 downregulated *NPPB* in cells that have a higher *MC5R* expression (Exp 1 and Exp 3), while an opposite effect was observed in cells that have a lower *MC5R* expression (Exp 2 and Exp 4).

Figure R4. Quantitative PCR analysis of *NPPB* mRNA expression in hiPSC-CMs treated with different concentrations of PG-901 for 24 hours in the absence (A) or presence (B) of ET-1 (100 nM). (C) Relative *MC5R* mRNA expression in unstimulated control cells in different experiments.

This could indicate that there is an off-target effect that becomes particularly dominant, when the signaling through MC5-R is weaker (*i.e.* in cells with lower *MC5R* expression). Therefore, in a model of advanced heart failure that was shown to be associated with reduced MC5-R dimerization (**Figure 3L**), this off-target effect might also become dominant and mask the potential anti-hypertrophic effect of MC5-R activation. This notion is further supported by the new *in vivo* experiment, which was performed to address the comment 3 of Referee #3. In sham-operated mice that

have a 'normal' cardiac MC5-R expression level, PG-901 induced a subtle reduction in ventricular weight at the higher dose (0.5 mg/kg) ($P=0.03$ vs Vehicle by Student's *t* test, $P=0.08$ by 1-way ANOVA and Dunnett post hoc test). We have discussed this issue also in the manuscript (page 14, lines 507-514)

4. MC5-R deletion from cardiomyocytes elevated cardiac hypertrophy under TAC. However, cardiac function was not affected. Is it due to the fact that control mice have not developed cardiac dysfunction yet? The TAC was too mild?

Response: We did not observe any change in LV ejection fraction (EF%) but the other functional indexes (FAC and IVRT) were reduced in response to TAC challenge, and Mc5r-cKO showed further reduction of FAC and IVRT, which indicates deterioration of systolic and diastolic performance. Fractional area change (FAC) measures the change in circumferential area of the LV and is thus analogous to LV ejection fraction (EF). Considering that EF is based upon (extrapolated) volume, FAC could be more sensitive than EF and detect subtle changes in LV systolic performance that are not necessarily captured by measuring EF. IVRT, on the other hand, is typically increased during the development of diastolic dysfunction, but in the case of restrictive filling (as in TAC-operated mice), IVRT declines as reported also in other mouse studies (e.g. Richards DA *et al*, *Sci Rep*. 2019 Apr 10;9(1):5844 & Sung MM *et al*, *Circ Heart Fail*. 2015 Jan;8(1):128-37).

Admittedly, as pointed out by the reviewer, control MYH6-MCM and Mc5r^{fl/fl} mice had not developed considerable cardiac dysfunction 4 weeks after the TAC operation. This mild phenotype might be caused by the genetic background (C57Bl/6J) of the (MYH6-MCM, Mc5r^{fl/fl} and Mc5r-cKO) mice. It has been systematically shown that the C57Bl/6J substrain is more resilient to TAC-induced heart failure and develop variable cardiac phenotypes compared to C57Bl/6N substrain (Garcia-Melendez L *et al*, *Am J Physiol Heart Circ Physiol*. 2013 Aug 1; 305(3): H397-H402 & Zi M *et al*, *Curr Res Physiol*. 2019 Dec;1:1-10). Four weeks after TAC, LV ejection fraction is marginally decreased and only a small percentage (~20%) of C57Bl/6J mice will eventually develop heart failure. These earlier observations are consistent with our data from MYH6-MCM, Mc5r^{fl/fl} and Mc5r-cKO mice (**Figure 6G**). This early time point (4 weeks after TAC) was selected to phenotype Mc5r-cKO mice, since advanced cardiac hypertrophy and heart failure was associated with reduced MC5-R expression in the heart (**Figure 3L**), which could limit the incremental effect of genetically-induced MC5-R deficiency on the hypertrophic response as discussed on manuscript page 15/lines 548-551.

5. Figure 8, the treatment of PG-901 was initiated since TAC. This is to test the prevention effect of α -MSH signaling. Could the authors try to test the rescue effect? For example, treat mice with PG-901 after the establishment of cardiac dysfunction (8 weeks after TAC).

Response: We thank the reviewer for this suggestion. It would be indeed valuable to test the rescue effect of PG-901 after the establishment of cardiac dysfunction.

However, PG-901 treatment was applied to TAC-operated C57Bl/6N mice that develop marked heart failure compared to C57Bl/6J (background strain for MYH6-MCM, Mc5r^{fl/fl} and Mc5r-cKO mice) and according to our experience, a significant number of C57Bl/6N mice would not survive beyond the 8-week time point. Furthermore, based on our data, it appears that PG-901 is not an optimal compound for selective targeting of MC5-R, because it does not precisely mimic the actions of α -MSH (*i.e.* no anti-hypertrophic effect *in vivo*) and might even evoke off-target effects (as discussed in response to comment 3). We would therefore prefer not to perform new *in vivo* experiments with PG-901 considering that the effects on EF and other functional parameters were only subtle at 5 weeks post-TAC. More detailed information on the pharmacokinetic and -dynamic characteristics of PG-901 would aid in designing future experiment with refined dosing regimens. Ideally, other selective MC5-R agonist could be tested and compared with PG-901 for their anti-hypertrophic and -fibrotic effects in TAC-operated mice.

Referee #3

This work presents a novel target involved in cardiac remodeling development. The authors show for the first time that α -MSH and the MC5R are expressed in mouse and human hearts. Most importantly, they report that MC5R is functionally active in ventricular myocytes. They also report that administration of a stable α -MSH analogue and an agonist for MC5R protects mice against TAC-induced cardiac remodeling. Furthermore, targeting specifically M5CR seems to improve cardiac function after TAC. The paper is interesting and the novelty of the results is high, nonetheless, there are some major points that need to be addressed.

Response: We thank the reviewer for the positive feedback as well as for the constructive comments.

Major concerns:

1. A major criticism is the organization of the results which makes difficult to the reader to understand what the authors want to show. The reviewer suggests to the authors to gather the in vitro parts.

Response: Thank you for pointing out the lack of clarity. Unfortunately, we weren't able to change the order of figures without strong disruptions to the logical flow of the manuscript, and dropping out significant amount of data. However, we have gone through each result section heading and updated the text where necessary to more carefully introduce the next topic.

2. Regarding the in vivo experiments, the number of animals is not the same in the different echocardiography parameters. For instance, LVPW (CTRL n= 11) and ejection fraction (CTRL n=9). If some animals are erased as outliers in any part of the work, they should be mentioned and considered for all the parameters. The number of animals used for each experiment and each condition should be detailed in the figure legends.

Response: We thank the reviewer for picking up this mistake. The graph for LVPW (Fig 1M) contained accidentally duplication of two data points. This has been corrected (->CTRL n=9) in the revised manuscript. This does not however affect the results/conclusion in any way as the difference between CTRL and TAC groups remains highly significant. We have also specified in more detail the number of animals used for each experiment in the figure legends.

3. The *in vivo* experiments with PG-901 or alpha-MSH should include SHAM mice treated with the drugs in order to show basal effects.

Response: As requested by the reviewer, we performed new *in vivo* experiments to investigate the effects of α -MSH and PG-901 in sham-operated mice. We observed that long-term treatment with α -MSH induced a subtle but significant reduction in ventricular weight and left ventricular posterior wall thickness in sham-operated mice without affecting functional parameters (Extended Figure 2). Likewise, high-dose of PG-901 tended ($P=0.03$ vs Vehicle by Student's t test, $P=0.08$ by 1-way ANOVA and Dunnett post hoc test) to reduce ventricular weight compared to vehicle-treated sham-operated mice (Appendix Table S2). Low- or high-dose treatment with PG-901 did not significantly affect systolic or diastolic performance of the heart as measured by echocardiography (Appendix Table S2).

4. All along the paper one of the most consistent results is the presence of fibrosis, therefore this should be measured for alpha-MSH treated mice (Figure 1H-N).

Response: We fully agree with the reviewer and quantified the level of LV fibrosis in α -MSH-treated mice. Consistent with the phenotype observed in Mc5r-cKO mice and PG-901-treated mice, TAC-induced cardiac fibrosis was significantly attenuated by chronic α -MSH treatment (Figure 1H & 1N).

5. The effect of the MC5-R agonist looks different from the rest (Figure 3A-D), but there is no statistical difference as there is for the MSH *in vitro*. It is difficult to observe the "clearly mimicked" stated by the authors. Also, it is important to consider that according to the paper cited by the authors (Grieco et al. 2002, Biochem. Biophys. Res. Commu, DOI:10.1006/bbrc.2002.6739), PG-901 behaves as a full agonist at the human MC5R but also as a full antagonist at MC3R and MC4R. Therefore, the authors should explain and discuss the rationale for choosing the PG-901 compound.

Response: We thank the reviewer for these remarks. We agree that the wording in that particular sentence was misleading and thus revised the text as follows: "...PG-901 was the only compound that showed similar responses to α -MSH". We have also included additional details on the pharmacological properties of PG-901 in the discussion (page 14/lines 507-514). The binding and antagonism of MC3-R and MC4-R might indeed explain some of the observed discrepancies related to PG-901. The rationale for choosing the PG-901 compound was based on previous publications as this appears to be the most widely used MC5-R selective agonist and also the only compound that has been previously used in *in vivo* studies (Trotta MC et al, Front

Physiol. 2018 Oct 26;9:1475. & Rossi S et al, *Mediators Inflamm.* 2016;2016:7368389). A few other MC5-R agonists have been developed and characterized but none of these are commercially available.

6. Upon treatment with PG-20N there seems to be a decrease in p-JNK and gene levels (Figures 3E-H). For instance, in the case of TGFB1 it is difficult to mention that MSH treated samples are different when treated with PG-20N. Authors should comment on this.

Response: Admittedly, this is something that has also puzzled us, since we expected that PG-20N alone would increase p-JNK and *Nppb* expression if anything. Nevertheless, the effect of PG-20N did not reach statistical significance in *post hoc* comparisons. Furthermore, in *post hoc* comparisons, there is a significant difference between Control/ α -MSH and PG-20N/ α -MSH treatments, further indicating that the effect of α -MSH is reversed by PG-20N. We have revised Figure 3E-H accordingly and displayed all the significant *post hoc* comparisons in the graphs.

7. Regarding inhibition of Gi with PTX, the results and description are both confusing. Authors state: " Further mechanistic experiments using H9c2 cells revealed that inhibition of Gi signaling with pertussis toxin (PTX) induced a further reduction in the amount of phosphorylated JNK and downregulation of Nppb and Acta2. Although PTX appeared to block the effect of PG-901 on JNK phosphorylation, it did not abrogate the gene expression changes evoked by MC5-R activation"

The comments are contradictory between the first and the second sentence. Following what was said before one would have expected an increase of the pJNK levels with PTX. The results show a reduction similar to that obtained with PG-901. Regarding gene expression, it seems there is a further decrease in some of them. It is rather difficult to draw a clear conclusion.

Response: We acknowledge the reviewer's concern and agree that the text in this regard is confusing. We have therefore revised the text as follows: "*Further mechanistic experiments using H9c2 cells revealed that inhibition of Gi signaling with pertussis toxin (PTX) induced a further reduction in the amount of phosphorylated JNK and downregulation of Nppb and Acta2. Importantly, it did not abrogate the gene expression changes evoked by MC5-R activation (Appendix Fig S4)*".

As elegantly pointed out by the reviewer, if Gi signaling mediates the reduction in p-JNK, then PTX would increase (rather than decrease) JNK phosphorylation. It might well be that the sensitivity of that particular ELISA assay was not sufficient to detect a further reduction in p-JNK level of PTX and PG-901-treated samples. Therefore, we focus on gene expression changes and particularly on *Nppb* expression, which serves as a molecular marker of cardiomyocyte hypertrophy. PTX did not reverse the downregulation of *Nppb* in PG-901 treated cells (**Appendix Figure S4**), indicating that Gi signaling does not mediate this effect. We also confirmed this result in primary cardiomyocytes (NMCs), which showed downregulation of *Nppb* in

response to PTX treatment (**Figure R5**). Importantly, the effect of PG-901 on *Nppb* expression was retained in PTX-treated NMCs (**Figure R5**).

Figure R5. Quantitative real-time PCR (qPCR) analysis of *Nppb* mRNA expression in NMCs treated with or without PTX for 18 hours followed by PG-901 treatment for 3 hours. Data are mean \pm SEM, $n=5-6$ per group in each graph. * $P<0.05$ for the indicated post hoc comparisons. ##### $P<0.0001$ for the main effect of PTX by 2-way ANOVA.

8. Figure 4J. There seems to be no increase in leucine incorporation after angiotensin II treatment in this experiment. Is this increase significant?

Response: This change is non-significant by 2-way ANOVA. According to our experience, Opti-MEM medium, which is used in combination with Lipofectamine, blunts the hypertrophic response to angiotensin II and other similar agonists.

9. Results in human samples are interesting but difficult to integrate with the data obtained in animals. The authors state that there is an increase in the MC5-R in the DCM/ICM patients when one would expect a reduction given the supposed beneficial effect of the receptor. Nonetheless, when they measured monomer and dimer of MC5R in mice they observed an increase in at least one of the forms. In line with this, the expression of POMC was also increased, when in the mouse TAC model the levels of alpha-MSH were clearly reduced. Yet, in the IPSC, contrary to what the authors observed in human samples, MC5R was reduced after stretching stress and ET-1 treatment. In the introduction, it is mentioned that a-MSH plasma levels are higher in heart failure patients than in control and in the discussion that plasma a-MSH levels are inversely correlated with NYHA functional class in heart failure. Controversially, after TAC in mice the levels of a-MSH were lower than in the control. The elevation or decrease of the a-MSH and MC5R levels in the heart and plasma in the context of a potential treatment should be further clarified and discussed.

Response: We acknowledge that these results might be confusing. However, the finding of differential expression of MC5-R in human cardiomyopathy samples is not necessarily contradictory, because there are multiple factors that might affect the outcome: e.g. mouse vs. human, protein vs. mRNA, age of subjects, etiology of disease and interfering signal from non-myocytes (in case of heart lysates). For example, it is not uncommon that mRNA level and corresponding protein product of a

gene are showing opposite effects, *i.e.* reduced protein level (due to enhanced degradation) could lead to a compensatory increase in mRNA level. Therefore, the direction of effect in the *MC5R* expression in human cardiomyopathy samples cannot be directly compared with *MC5R* protein expression in the mouse heart after TAC-induced cardiac hypertrophy. Furthermore, the levels of α -MSH in the plasma/heart cannot be directly compared between human cardiomyopathy patients and TAC-challenged mice. We observed that there was a biphasic response to TAC in terms of cardiac *Pomc* expression, *i.e.* increased expression during the early phase after TAC and then declining expression towards advanced heart failure (**Figure 1E and F**). This fits with the clinical finding of higher plasma α -MSH concentration in patients with hypertrophic cardiomyopathy (vs dilated cardiomyopathy) and with the negative association between plasma α -MSH concentration and NYHA functional class, *i.e.* declining α -MSH level in more advanced heart failure (*Yamaoka-Tojo M et al., Intern Med. 2006;45:429-434*).

However, as pointed out by the reviewer, the downregulation of *MC5R* in stretched and ET-1-treated hiPSC-CMs is conflicting with the increased *MC5R* in DCM and ICM patient samples. This discrepancy might relate to the handling of the control and patient samples: LV samples from DCM and ICM patients were collected immediately upon heart cardiac transplantation, while control LV samples were collected at autopsy. Although control and patient samples showed stable and consistent Ct values for the housekeeping gene, the difference in sample handling (*i.e.* delay in freezing of control LV samples) might introduce a technical error in quantification of certain target genes. Therefore, a more reliable and important finding in human heart samples is the highly significant correlation between *POMC* and *MC5R* expression in both DCM and ICM groups.

If the reviewers and editors feel that these results are conflicting and confusing for the readers, we could exclude the data from human cardiomyopathy samples (**Figure 5A-C**) and related text parts completely from the manuscript.

10. For the TAC model in the cardiac-specific mice there are several points to review. Statistical differences between CTRL and CTRL TAC should be shown in all the parameters, to see if the model works, main effect is not sufficient when one of the conditions is the treated one, post-hoc test should be displayed. In addition, the differences between the KO and the 2 controls are not in all cases significant, making the result difficult to interpret, please comment on this. On the other hand, the TAC model (4 weeks) seems to have no effect on cardiac function, taking into consideration the TAC model used in Figure one (8 weeks) where the functional effect is obvious, The reviewer does not understand why this experiment was performed at 4 weeks and not 8 weeks. Also, the authors add endocardial FAC and IVRT analysis, which they do not analyze in the other case, and is only different from one of the control groups. Finally, the authors discuss the tamoxifen in the discussion section, this should be described in the results and the low ejection fraction in the SHAM group should also be discussed.

Response: We have carefully checked the figures (**Fig 1** and **Fig 8**) and included missing statistical differences between CTRL and CTRL TAC by 1-way ANOVA and *post hoc* tests. In Figure 6 and 7, we argue for displaying the main effect of TAC by 2-way ANOVA, because GraphPad Prism 9 does not allow analyzing pre-defined comparison in *post hoc* tests of 2-way ANOVA. If we choose to analyze and display all the possible *post hoc* comparisons, we will end up with 15 different comparisons making the graphs extremely busy and the interpretation whether the TAC model works or not even more obscure for the reader. Two-way ANOVA is a valid statistical test to analyze the effect of two independent variables (*i.e.* TAC and genotype). We have modified the presentation of statistical significances in **Fig 6** and **7** to make it clearer whether TAC has an effect on the analyzed parameters.

It is true that the differences between the Mc5r cKO and the 2 controls (Mc5r^{fl/fl} and MYH6-MCM) are not in all cases significant. However, the main findings, which are increased ventricular size and cardiac fibrosis, are significant in comparisons against the 2 controls.

In **Fig 6** and **7**, we chose to phenotype Mc5r-cKO mice already at 4 weeks post-TAC, because advanced cardiac hypertrophy and heart failure was associated with reduced MC5-R expression in the heart (**Figure 3L**), which could limit the incremental effect of genetically-induced MC5-R deficiency on the hypertrophic response (as discussed on manuscript **page 15/lines 548-551**). It has been systematically shown that the C57Bl/6J substrain (background strain of Mc5r-cKO mice) is more resilient to TAC-induced heart failure and develop variable cardiac phenotypes compared to C57Bl/6N substrain (*Garcia-Melendez L et al., Am J Physiol Heart Circ Physiol. 2013 Aug 1; 305(3): H397-H402 & Zi M et al., Curr Res Physiol. 2019 Dec;1:1-10*). Four weeks after TAC, LV ejection fraction is marginally decreased and only a small percentage (~20%) of C57Bl/6J mice will eventually develop heart failure. These earlier observations are consistent with our data from MYH6-MCM, Mc5r^{fl/fl} and Mc5r-cKO mice. Since LV EF, as a primary functional parameter, was not changed in Mc5r-cKO at 4-weeks post-TAC, we did a more extensive analysis by echocardiography to detect more subtle changes in LV systolic performance. Fractional area change (FAC) measures the change in circumferential area of the LV and is thus analogous to LV ejection fraction (EF). Considering that EF is based upon (extrapolated) volume, FAC could be more sensitive than EF and detect changes in LV systolic function that are not necessarily captured by measuring EF. IVRT was analyzed as a part of evaluation of diastolic function (Figure 6&8), which was included in the echocardiography protocol based on the *in vitro* findings that α -MSH and PG-901 down-regulated fibrotic markers, which might have an influence on myocardial stiffness and diastolic function *in vivo*. Our ultrasound system was also upgraded with tools for pulsed-wave and tissue Doppler imaging after we had already completed the *in vivo* experiment for Figure 1.

As requested by the reviewer, we moved the discussion of the tamoxifen-effect to the results section (**page 11/lines 369-375**). Ejection fraction was consistently low in

sham-operated mice between different experiments (Figure 1, 6 & 8). The EF values are also comparable to our previous publications (*Perjés Á et al., Basic Res Cardiol. 2016 Jan;111(1):2* & *Szabó Z et al., Hypertension. 2014 Jun;63(6):1235-40*) and thereby more dependent on the anesthesia protocol than on tamoxifen-related cardiotoxicity.

11. In the last experiment with PG-901 there seems to be some relevant functional data. Nonetheless, cardiac hypertrophy is not modified by the treatment. This might be due to the off-target effect of the drug which could explain the absence of any effect at higher doses. This should be discussed in light of a potential treatment and further clarified.

Response: This is an important remark that was also brought up by Referee #2. Admittedly, there is something with PG-901 that we have not figured out. For example, in hiPSC-CMs, PG-901 evokes a significant and dose-dependent effect on *NPPB* expression, but the direction of effect varies from experiment to experiment (**Figure R4**). Basal *MC5R* expression, which varies between different batches of hiPSC-CM differentiation, might explain this confusing finding. PG-901 downregulated *NPPB* in cells that have a higher *MC5R* expression (Exp 1 and Exp 3), while an opposite effect was observed in cells that have a lower *MC5R* expression (Exp 2 and Exp 4).

Figure R4. Quantitative PCR analysis of *NPPB* mRNA expression in hiPSC-CMs treated with different concentrations of PG-901 for 24 hours in the absence (A) or presence (B) of ET-1 (100 nM). (C) Relative *MC5R* mRNA expression in unstimulated control cells in different experiments

This could indicate that there is an off-target effect that becomes particularly dominant, when the signaling through MC5-R is weaker (*i.e.* in cells with lower *MC5R* expression). Therefore, in a model of advanced heart failure that was shown to be associated with reduced MC5-R dimerization (**Figure 3L**), this off-target effect might also become dominant and mask the potential anti-hypertrophic effect of MC5-R activation. This notion is further supported by the new *in vivo* experiment, which was performed to address the comment 3. In sham-operated mice that have a ‘normal’ cardiac MC5-R expression level, PG-901 induced a subtle reduction in ventricular weight at the higher dose (0.5 mg/kg) ($P=0.03$ vs Vehicle by Student’s t test, $P=0.08$ by 1-way ANOVA and Dunnett post hoc test).

Furthermore, as a part of more thorough phenotyping, we quantified total leukocytes and their different subsets in the blood of PG-901-treated TAC mice (**Figure R6**), since MC5-R is known to regulate immune responses in leukocytes. This immunophenotyping revealed that PG-901 inhibits the TAC-induced increase in circulating neutrophils (**Figure R6B**) and decrease in B cells (**Figure R6D**) in a dose-dependent manner, *i.e.* the higher dose of PG-901 was more effective than the lower dose. Consequently, the possible off-target effect appears to be heart-specific, since PG-901 showed classical dose-responsiveness in other cells. The issue related to off-target effect of PG-901 is discussed on manuscript **page 14/lines 507-514**.

Figure R6. . Quantification of total leukocytes (A), neutrophils (B), monocytes (C) and B cells (D) in the peripheral blood of sham- and TAC-operated mice treated with either vehicle or PG-901. Data are mean \pm SEM, each dot represents individual mouse. * $P < 0.05$ for the indicated comparisons by 1-way ANOVA and Dunnett post hoc tests.

12. The work is lacking a mechanism analysis that could explain the obtained results. Though the JNK pathway seems plausible, there are no measurements of JNK in any of the animal models used. Even the authors discard GS and GI pathways, for cAMP there are different effectors from PKA that could be playing a role and other JNK partners should be evaluated. Also, given the role of JNK in apoptosis, this should be evaluated both *in vitro* and *in vivo*.

Response: Our *in vitro* data show that the MC5-R-mediated effects on hypertrophy- and fibrosis-associated markers are dependent on JNK signaling (**Figure 3L-N**). In the revision, this dependency was also confirmed to exist in primary cardiomyocytes (**Extended Figure 4C**). We also did a further screening of possible targets that are known to be modulated by MC5-R and other MC-R subtypes. However, none of the other intracellular targets were affected by MC5-R signaling (**Figure R1** and **Extended Figure 4A and B**). For further details, please see also response to comment 1 of Referee #1 (page 2 and 3 of this response letter).

We also analyzed p-JNK level in the heart of Mc5-cKO and PG-901-treated mice, but we did not find any significant changes in this regard (for further details and the results, please see response to comment 2 by Referee #1). Based on the existing and conflicting literature on the role of JNK in cardiac hypertrophy (reviewed in *Physiol Rev.* 2010 Oct;90(4):10.1152/physrev.00054.2009), it would be highly challenging to prove the contribution of the JNK pathway to the observed phenotypes *in vivo*. Earlier

findings indicate that JNK activation is a dynamic and transient signaling event (peaking at ~30 min) after applying pressure overload and that different JNK isoforms might have separate and non-redundant roles in the process. Although *in vitro* studies strongly argue for a prohypertrophic role of JNKs, loss-of-function approaches to silence JNK or its upstream regulators *in vivo* have resulted in promotion as well as attenuation of cardiac growth response (as discussed on manuscript **page 15, lines 531-535**). Furthermore, JNK activation *in vivo* (in transgenic animal models) does not induce cardiac hypertrophy.

Therefore, we have been careful not to claim that the observed *in vivo* phenotypes would be dependent on the JNK pathway, and disclose the uncertainty related to the involvement of JNK signaling (manuscript **page 15, lines 535-539**): “*In vitro* experiments in the current study demonstrate that MC5-R regulates hypertrophic growth of cardiomyocytes in a JNK-dependent manner. However, further studies are warranted to determine whether MC5-R-induced JNK reduction has a direct effect on cardiomyocyte growth *in vivo*”.

As requested by the reviewer, we analyzed markers of apoptosis in *Mc5r*-silenced NMCs. MC5-R deficiency was associated with increased expression of pro-apoptotic markers such as *Casp3*, *p53*, *Bax* and *Noxa* (**Extended Figure 4D-H**). In good agreement with this finding, TAC-operated *Mc5r*-cKO mice showed increased number of apoptotic TUNEL⁺ cells in the heart (**Appendix Figure S9C**). Conversely, long-term treatment with PG-901 reduced the number of TUNEL⁺ cells in the heart (**Extended Figure 5**). Given the established role of JNK in apoptosis, it was surprising that the induction of apoptotic markers in *Mc5r*-silenced NMCs occurred in a JNK-independent manner (**Extended Figure 4D-H**). In the screening of other targets, there was a tendency that p38 phosphorylation ($P=0.14$ vs Control siRNA) could be also affected in *Mc5r*-silenced cells (**Extended Figure 4A and B**). Therefore, we also investigated whether p38 inhibition with TAK-715 could reverse the upregulation of the pro-apoptotic markers in *Mc5r*-silenced NMCs. TAK-715 alone increased the expression of *Nppb* (**Figure R7A**), *Casp 3* (**B**) and *Noxa* (**C**), and did not reverse the effect of *Mc5r* knockdown on these genes, thus excluding the possibility that the increased apoptosis is mediated by p38 signaling.

Figure R7. Quantitative real-time PCR (qPCR) analysis *Nppb*, *Casp3* and *Noxa* mRNA expression in NMCs treated with or without the JNK inhibitor SP600125 (10 μ M) or the p38 inhibitor TAK-715 (5 μ M) for 30 minutes followed by transfection with control siRNA or Mc5r targeting siRNA for 24 hours. Data are mean \pm SEM, n=6 per group in each graph. * $P < 0.05$, ** $P < 0.01$, *** $P < 0.001$ and **** $P < 0.0001$ for the indicated post hoc comparisons by 2-way ANOVA and Bonferroni post hoc tests.

13. In the discussion authors state:

"fibroblasts are considered as important effector cells in the hypertrophic response to Ang II and Mc5r-cKO mice were not sensitized to Ang II-induced cardiac hypertrophy. Furthermore, Ang II infusion did not change MC5-R expression in the heart, while TAC surgery clearly affected cardiac MC5-R protein levels. These findings suggest that MC5-R signaling does not modulate hypertrophic remodeling that is primarily driven by cardiac fibroblasts or other non-myocytes."

This does not correlate with the results in vitro where PG-901 is able to reduce AngII prohypertrophic effects. This should be clarified.

Response: We do not fully understand the point raised by the reviewer. We speculate that the difference (*in vitro* vs *in vivo*) might be caused by the dominant influence of noncardiomyocyte cells such as fibroblasts as mediators of Ang II-induced hypertrophic response *in vivo*. Vasoconstriction and consequent increase in blood pressure also contribute to cardiac hypertrophy in Ang II-infused mice. Accordingly, the mechanisms of Ang II-induced cardiomyocyte hypertrophy differ between *in vitro* and *in vivo* settings. This could explain why MC5-R activation/inactivation evokes a clear phenotype (in the context of Ang II stimulation) only in cultured cardiomyocytes, when the cells are isolated from the influence of other cell types such as fibroblasts.

Minor points

1. The number of positive POMC cells and triple positive are rather low. Given the immunohistochemistry image, the reviewer would expect a higher number of cells. The authors should discuss this point.

Response: The number of *Pomc*⁺ and *Pomc*⁺ *Cpe*⁺ *Pam*⁺ -cells was relatively low and underestimated due to the low detection sensitivity of scRNA-seq technology and consequent dropout of low-expression genes such as *Pomc*. This is now mentioned also in the manuscript (**page 6/lines 165-168**).

2. The western blot in Sup Fig 1G seems to show the reduction the authors state, but the bands are not clear enough, especially the GAPDH.

Response: Based on the reviewer's concern, we rerun the samples and replaced the the representative blots with these new images to improve the quality of the bands (**Extended Figure 1F**).

3. Although it makes sense to treat mice with the alpha-MSH after two weeks of TAC, the logic behind this design and the selected dose should be better described.

Response: The primary reason for this design was to allow mice a sufficient recovery period after TAC surgery and to let a certain degree of cardiac hypertrophy to take place before starting the treatment. The dose selection was based on our earlier studies, which demonstrated that 0.3 mg/kg dose was safe and therapeutically effective in other disease models such as atherosclerosis (*Rinne P et al, Arterioscler Thromb Vasc Biol. 2014 Jul;34(7):1346-54* & *Rinne P et al, Cardiovasc Res. 2013 Feb 1;97(2):360-8*). These details have been added to the revised manuscript (**page 17/lines 597-599**).

4. Figure 1H-L number and localization of cardiomyocytes quantified for the experiments should be detailed.

Response: This information has been included in the revised manuscript (**page 22/lines 784-787**): “*For cell size quantification, at least 100 individual cells with well-defined cell membranes and visible cell nuclei were selected and measured in fields of longitudinally oriented cardiomyocytes*”.

5. Sup Fig II individual points should be added to the graphs.

Response: The figure has been modified as requested (**Appendix Figure S1**).

6. Figure 2 K-M. How long was the treatment of Angiotensin for qPCR analysis? One day like for hypertrophy measurement?

Response: We thank the reviewer for picking up this missing information. For qPCR analysis, the cells were treated with Ang II (or other hypertrophic agents) for 3 hours

(Figure 2K-M) like in 4F-G, as the gene expression changes occurred quickly after applying the treatment in these cells.

7. Throughout the paper the loading control for the western blot changes arbitrarily. This should be homogenized or explained.

Response: For revision, we have limited the use of loading controls to β -actin and vinculin, and excluded GAPDH that was originally used in **Extended Figure 1F**. For an unknown reason, our protocol for stripping and re-probing of WB membranes has led to inconsistent results in terms of wiping out the primary antibody. Therefore, in few cases, we needed to use vinculin (as a loading control) that is a higher MW protein, which signal did not overlap with the signal from the primary protein target.

8. Protein levels of MC5R in the cardiac-specific KO are not convincing given the unconvincing signal of the Vinculin.

Response: Based on the reviewer's concern, we rerun the samples and replaced the representative blots with these new images to improve the quality of the bands (**Appendix Figure S7C**).

9. In human iPSC cells the reviewer does not understand the point of analyzing ET1 when it was already discarded in figure 4 for not showing a response to MC5R agonist and JNK pathway.

Response: This is a relevant point that need to be explained also in the manuscript (**page 10/lines 342-344**). Previous publications have revealed that angiotensin II or α -adrenoceptor agonists do not evoke hypertrophic responses in hiPSC-CMs (Földes G et al., *Stem Cell Reports*. 2014 Nov 11;3(5):905-14). There is clearly differences between cell models (e.g. NMCM vs hiPSC-CM) how they respond to hypertrophic stimulation. This was the reason why we used ET-1 as a hypertrophic agent in hiPSC-CM experiments.

10. There are some writing mistakes in the text and the figures. This should be checked.

Response: We have now carefully proofread the manuscript and hopefully found and corrected all the writing mistakes.

11. The paper would profit from a graphical abstract

Response: We thank the reviewer for this suggestion and have drafted a graphical abstract to summarize the key findings of the study.

Dear Dr. Rinne,

Thank you for transferring your manuscript, which was previously revised and re-reviewed at another journal. As my colleague mentioned in her letter to you, we would like to offer publication pending satisfactory revision as per the concerns of referee #3 regarding the statistical analyses (points 2-8). Please provide a complete point-by-point response to the referee reports.

Moreover, I need you to address the points below.

- Please update the Author Checklist according to the journal name (EMBO Reports) and the manuscript number.
- Please remove the 'Author contributions' section from the manuscript.
- Please rename the 'Conflict of interest' section as 'Disclosure and competing interests statement'.
- Please remove the 'The paper explained' section from the manuscript, which does not suit the format of EMBO Reports.
- We note that some references have DOIs, which are only needed for preprints and datasets that have not been published yet. Please remove the DOIs of the published references.
- The figure and table names need correcting in the Table of Contents of the Appendix file by adding "Appendix" to each title (e.g. Table S1 should be Appendix Table S1).
- We note that the provided source data are currently incomplete. Please submit complete source data as per the Source Data checklist provided by our Source Data coordinator Dr. Hannah Sonntag. Please submit them as one file per figure (please see <https://www.embopress.org/pb-assets/embo-site/Guide%20for%20SourceData%20Submission-1656066810500.pdf>).
- The synopsis image needs to be 550px wide and 300-600px high. When your synopsis image is resized accordingly, the labels are too small to read. Please increase the font size of the synopsis image labels.
- We note that the following figure panels are currently not called out in the text: Fig 1K, Fig 2G, Fig 4K, Fig EV5.
- The legends for EV figures need correcting to "Expanded View Figure Legends" in the heading and "Figure EV1" etc. for the figure titles.
- Our production/data editors have asked you to clarify several points in the figure legends:
Figure legends:
 1. Please note that the 'Data Information' section is not labelled in the legends of figures 1; 2; 3; 4; EV 2a-f; EV 3; EV 4.
 2. Please note that a separate 'Data Information' section is required in the legends of figures 8b-h.
 3. Please note that in the legend of figure 8, the p value is provided however asterisk "*****" for the same is not mentioned. This needs to be rectified.
 4. Please note that the figures 3i-j does not contain any quantification graph, kindly rectify the statistical test related information in the figure legend appropriately.
 5. Please define the annotated p value * in the legend of figure 5d, as appropriate.
 6. Please indicate the statistical test used for data analysis in the legends of figures 4a, c-d; 5c.
 7. Please note that in figures 4l, n; 5f-g; 6e; there is a mismatch between the annotated p values in the figure legend and the annotated p values in the figure file that should be corrected.
 8. Please note that the error bars are not defined in the legends of figures EV 1f-i.

Thank you again for giving us to consider your manuscript for EMBO Reports, I look forward to your minor revision.

Kind regards,

Deniz Senyilmaz Tiebe

--

Deniz Senyilmaz Tiebe, PhD
Scientific Editor
EMBO Reports

EMBOR-2024-58845-T

Title: α -Melanocyte-stimulating hormone regulates pathological cardiac remodeling via melanocortin 5 receptor

Corresponding author: Dr. Petteri Rinne

Authors' response to reviewers and editors

Referee #1:

The revised version of the manuscript is clearly improved and the authors adequately responded to the majority of the points I have raised. However, they still do not present a convincing signaling mechanism downstream of MC5R. They should really conduct some unbiased phosphoproteomics to find out how α -MSH acts on cardiomyocytes. Alternatively, the authors could investigate whether PKA downstream of cAMP might mediate the effects. In addition, I do not understand, why the authors could not measure α -MSH or maybe Pomc mRNA in the LV samples of ICM and DCM patients, which they investigated in Figure 5.

Response: We thank the reviewer for the comments. We had already performed the alternative suggestion of the reviewer and demonstrated that PKA inhibition with H89 does not reverse the effects of MC5-R activation on p-JNK level or gene expression in cultured cardiomyocytes (**Appendix Figure S4**). Performing phosphoproteomics might give us some new ideas of the intracellular signaling mechanism, which we could further test in mechanistic experiments *in vitro* in a similar fashion as we have already done in the case of JNK pathway. Nevertheless, these additional experiments would not lead to any new significant conclusions. The main mechanism, which we identified in this study, is that α -MSH regulates pathological cardiac remodeling by activating MC5-R in cardiomyocytes.

α -MSH is derived from posttranslational cleavage of the precursor molecule POMC and thus, cannot be measured at the mRNA level. *POMC* mRNA level was already measured in DCM and ICM patient samples, and the results are reported in **Figure 5B** and **5C**.

Referee #3:

The authors have invested great efforts to address comments raised by the referees and have significantly improved the manuscript. However, the study still lacks mechanistical insights into MC5-R biological action in the context of cardiac remodeling. The observation that MC5-R activation protects against cardiac remodeling is of potential

medical value. However, it should be demonstrated how does MCR5 activation prevent remodeling. In addition, there are still some discrepancies between the different in vitro and in vivo models. Of particular importance, the stat analysis should be revised as suggested below to the authors.

The authors did a great work answering the questions raised. However, some important points still need to be clarified.

1. The paper still lacks a thorough mechanistical analysis to explain how MC5-R activation prevents cardiac remodeling.

Response: We thank the reviewer for the comments. The main mechanism, which we identified in this study, is that α -MSH regulates pathological cardiac remodeling by activating MC5-R in cardiomyocytes. MC5-R activation directly and immediately evokes anti-hypertrophic changes at the level of gene expression and protein synthesis. If the reviewer refers to the intracellular signaling mechanism with this comment, we have demonstrated that the effects of MC5-R on the hypertrophic response *in vitro* are reversed by JNK inhibition. We could perform additional mechanistic experiment to identify additional nuances in the signaling response downstream of MC5-R, but the main conclusions of the manuscript would still remain the same. As disclosed in the previous response letter, we cannot obtain new *in vivo* evidence in a reasonable time frame to support the role of JNK or some other intracellular signaling pathway in the observed *in vivo* phenotypes.

2. The authors have added the number of experiments and technical replicates for each graph. In the case of the qPCR analysis many times there are just two experiments and 4 technical replicates. It is not correct to graph technical replicates in this kind of analysis, especially when the dispersion between the technical replicates is big. It would be necessary to show the mean of at least three independent experiments.

Response: In a strict sense, the qPCR data points represent technical replicates. The individual data points are derived from cells that were grown on separate plates, but these cells/wells were treated within the same day and therefore should be regarded as technical replicates. Therefore, the variation between individual data point is higher than would be expected from pure technical replication. The use of these kind of technical replicates appears to be also acceptable in recently published *EMBO Mol Med* and *EMBO Reports* papers:

<https://www.embopress.org/doi/full/10.15252/emmm.202317601>

<https://www.embopress.org/doi/full/10.15252/embr.202357600>

The reproducibility of the main *in vitro* effects is demonstrated by a series of individual experiments that were carried out to investigate the time- and dose-dependency of the effects of α -MSH/PG-901 as well as the reversal of the observed effects in multiple mechanistic experiments. Furthermore, we demonstrated that the

effects are appearing similarly in H9c2 cells and NCMs. In experiments with hiPSC-CMs, it is easier to recognize and utilize true biological replicates, which we have consistently used with this cell model.

3. Response to Point 10 : *"Statistical differences between CTRL and CTRL TAC should be shown in all the parameters, to see if the model works, the main effect is not sufficient when one of the conditions is the treated one, the post-hoc test should be displayed. ..."*

The answer is not sufficient. Authors should show the posthoc test in a table or the text. The main effect difference indicates that there are global differences between SHAM and TAC but not specifically between controls. This statistical analysis needs to be shown to convincingly demonstrate that results of the experiments are in line with conclusions made by the authors.

Response: We have presented the significant *post hoc* comparison in the graphs (Figure 6&7). None of the *post hoc* comparisons between the two control groups (Mc5r^{fl/fl} and MYH6-MCM) within sham- or TAC-operated mice were significant. We would highly appreciate if the reviewer could indicate the exact lines in the manuscript text where we have stated an effect or reported a result that is not supported by an appropriate statistical test.

4. The authors did not answer to the Reviewer 3 comment on the absence of any effect of PG-20N on TGFBI and acta2. Please revise the following sentence and mitigate the conclusion. 248-250: *"Addition of the selective MC5-R antagonist PG-20N completely reversed the effect of α -MSH on p-JNK level and gene expression changes (Fig 3E-H), further supporting the role of MC5-R as a mediator of the α -MSH-induced effects."*

Response: We have revised the sentence as requested by the reviewer.

5. Figure 5H. Please clearly indicate the number of experiments and replicates. The graph quantification of immunoblot should be presented with columns.

Response: The number of experiments is indicated in the corresponding figure legends (n=3 biological replicates from 3 independent experiments). Due to high variation in the magnitude of the response to ET-1 stimulation between individual experiments, we argue that it is more informative and transparent to show the trends in the response to PG-901 within each individual experiment.

6. Response to Point 8. Again, it is not clear to Reviewer 3 why the increase is mentioned significant in Figure 4K (Figure 4J in the first version of the MS).

Response: We would appreciate if the reviewer could point the exact lines in the manuscript where we state that the effect of Ang II is significant. The only significance was observed for *Mc5r* knockdown as indicated in the text and in the Figure 4K. Since there was no effect for Ang II, we modified the graph and removed the data of Ang II-stimulated Ctrl siRNA and Mc5r siRNA samples (Fig 4K).

7. Response to Point 9. I would not withdraw the human data. A paragraph dedicated to the discrepancies between the different models (mice, hiPS, human samples) used in the study should be presented in the discussion.

Response: We thank the reviewer for this insight. We have discussed the discrepancies between the different models on manuscript **page 13/lines 505-509**.

8. Response to Point 11. The graphical abstract is missing in the revised version of the MS.

Response: The graphical abstract was provided as a separate file named as 'Cover art/synopsis image'

Editorial comments:

Please update the Author Checklist according to the journal name (EMBO Reports) and the manuscript number.

Response: Revised as requested.

Please remove the 'Author contributions' section from the manuscript.

Response: Revised as requested.

Please rename the 'Conflict of interest' section as 'Disclosure and competing interests statement'.

Response: Revised as requested.

Please remove the 'The paper explained' section from the manuscript, which does not suit the format of EMBO Reports.

Response: We removed the 'The paper explained' section from the manuscript.

We note that some references have DOIs, which are only needed for preprints and datasets that have not been published yet. Please remove the DOIs of the published references.

Response: We removed all the DOIs from the published references.

The figure and table names need correcting in the Table of Contents of the Appendix file by adding "Appendix" to each title (e.g. Table S1 should be Appendix Table S1).

Response: Revised as requested.

We note that the provided source data are currently incomplete. Please submit complete source data as per the Source Data checklist provided by our Source Data coordinator Dr. Hannah Sonntag. Please submit them as one file per figure (please see

<https://www.embopress.org/pb-assets/embo-site/Guide%20for%20SourceData%20Submission-1656066810500.pdf>

Response: We submitted complete source data as instructed by Dr. Hanna Sonntag.

The synopsis image needs to be 550px wide and 300-600px high. When your synopsis image is resized accordingly, the labels are too small to read. Please increase the font size of the synopsis image labels.

Response: We increased the font size of the synopsis image labels as requested.

We note that the following figure panels are currently not called out in the text: Fig 1K, Fig 2G, Fig 4K, Fig EV5.

Response: We revised the manuscript text to add the missing figure references.

The legends for EV figures need correcting to "Expanded View Figure Legends" in the heading and "Figure EV1" etc. for the figure titles.

Response: Revised as requested.

Our production/data editors have asked you to clarify several points in the figure legends:

Figure legends:

1. Please note that the 'Data Information' section is not labelled in the legends of figures 1; 2; 3; 4; EV 2a-f; EV 3; EV 4.

Response: We added the missing 'Data information' sections to the manuscript.

2. Please note that a separate 'Data Information' section is required in the legends of figures 8b-h.

Response: Revised as requested.

*3. Please note that in the legend of figure 8, the p value is provided however asterisk "****" for the same is not mentioned. This needs to be rectified.*

Response: We added the missing "****" -annotation to the figure legend.

4. Please note that the figures 3i-j does not contain any quantification graph, kindly rectify the statistical test related information in the figure legend appropriately.

Response: Revised as requested.

*5. Please define the annotated p value * in the legend of figure 5d, as appropriate.*

Response: Revised as requested.

6. Please indicate the statistical test used for data analysis in the legends of figures 4a, c-d; 5c.

Response: The missing information was added to the figure legends.

7. Please note that in figures 4l, n; 5f-g; 6e; there is a mismatch between the annotated p values in the figure legend and the annotated p values in the figure file that should be corrected.'

Response: We revised the figure legends to include the missing p values.

8. Please note that the error bars are not defined in the legends of figures EV 1f-i.

Response: We added the missing information to the figure legend.

Dear Dr. Rinne,

Thank you for submitting your revised manuscript. I have now looked at everything and all is fine. Therefore, I am very pleased to accept your manuscript for publication in EMBO Reports.

Before exporting your manuscript to our production team, I need your input on one point. I propose a minor change in the title to increase clarity/accessibility of the findings. Please let me know if you approve this change.

The proposed title: α -Melanocyte-stimulating hormone alleviates pathological cardiac remodeling via melanocortin 5 receptor

Congratulations on a nice work!

Kind regards,

Deniz Senyilmaz Tiebe

--

Deniz Senyilmaz Tiebe, PhD
Editor
EMBO Reports

--
